# Learning Equivalence Classes of Bayesian Network Structures with GFlowNet

**Michelle Liu**  *2michelle.liu@gmail.com*
*Mila – Québec AI Institute*

**Zhaocheng Zhu**  *zhaocheng.zhu@mila.quebec*
*Mila – Québec AI Institute*

**Olexa Bilaniuk**  *olexa.bilaniuk@mila.quebec*
*Mila – Québec AI Institute*

**Emmanuel Bengio**  *emmanuel.bengio@valencelabs.com*
*Valence Labs*

**Reviewed on OpenReview:** *https://openreview.net/forum?id=FAcc7oAdaa*

## Abstract

Understanding the causal graph underlying a system is essential for enabling causal inference, particularly in fields such as medicine and genetics. Identifying a causal Directed Acyclic Graph (DAG) from observational data alone is challenging because multiple DAGs can encode the same set of conditional independencies. These equivalent DAGs form a Markov Equivalence Class (MEC), which is represented by a Completed Partially Directed Acyclic Graph (CPDAG). Effectively approximating the CPDAG is crucial because it facilitates narrowing down the set of possible causal graphs underlying the data. We introduce CPDAG-GFN, a novel approach that uses a Generative Flow Network (GFlowNet) to learn a posterior distribution over CPDAGs. From this distribution, we sample high-reward CPDAG candidates that approximate the ground truth, with rewards determined by a score function that quantifies how well each graph fits the data. Additionally, CPDAG-GFN incorporates a sparsity-preferring filter to enhance the set of CPDAG candidates and improve their alignment with the ground truth. Experimental results on both simulated and real-world datasets demonstrate that CPDAG-GFN performs competitively with established methods for learning CPDAG candidates from observational data.

## 1 Introduction

Causal graphs produced from observational data are highly sought after, because knowing the causal dag underlying a system enables counterfactual reasoning, allows predictions about the system, and may enhance the generalizability of machine learning models (Schölkopf et al., 2021).

A causal structure is typically represented by a Directed Acyclic Graph (DAG). However, a significant challenge arises because it is often impossible to determine which DAG represents the true causal structure from observational data alone. This occurs because multiple DAGs can encode the same set of conditional independencies, making them equally valid representations of the causal structure. Consequently, DAGs that encode the same set of conditional independencies can be grouped into a single class known as a Markov Equivalence Class (MEC) (Castelletti et al., 2018). Thus, from observational data alone, we can learn a causal graph only up to its MEC (Chickering, 2002a; Koller & Friedman, 2009)[1].

---

[1]While it is commonly the case that DAGs can be learned only up to their Markov Equivalence Class (MEC) from observational data alone, there are special cases under certain conditions where exact identification is possible. For detailed examples, see Shimizu et al. (2006); Hoyer et al. (2008); Peters & Bühlmann (2014).

A MEC is represented by a Completed Partially Directed Acyclic Graph (CPDAG, Castelletti et al., 2018, see Figure 1). Popular methods, such as the PC algorithm (Spirtes et al., 2001), typically identify only a single CPDAG from observational data, potentially overlooking other promising candidates. Like all algorithms, these methods rely on certain assumptions for optimal performance, some of which are challenging to meet in practice. For instance, the PC algorithm often requires unrealistic conditions, such as infinite data or perfect oracles for independence tests - conditions that are difficult to meet in practice. Consequently, these methods may fail to accurately identify the true CPDAG. If the true causal structure falls into a different equivalence class than the one predicted by the model, sticking to one class may overlook potential better-fitting models. Given this limitation, an algorithm that returns multiple CPDAGs from observational data is preferable.

One approach is to adopt a Bayesian method to obtain a posterior distribution over all possible CPDAGs, allowing for the sampling of multiple CPDAGs that could explain the data. In this spirit, we propose to combine a Bayesian approach and a filtering mechanism. While our approach shares a connection to the GFlowNet framework, it is fundamentally different from methods like DAG-GFN [2].

Deleu et al. (2022) introduce a novel approach called DAG-GFN, which uses GFlowNet with a uniform prior to approximate the posterior distribution over DAGs from observational data, where probabilities are approximately proportional to the reward [3]. The reward is determined by a score function that measures how well a DAG fits the observations. The better the fit, the higher the probability of sampling that DAG. CPDAGs can be obtained by converting the sampled DAGs to CPDAGs.

One drawback that stems from using this approach is primarily due to its reliance on a score function and a uniform prior. Since the posterior distribution trained by DAG-GFN is approximately proportional to the density induced by the score function, the learned distribution is heavily influenced by this function. If the score function assigns high scores to many DAGs, including those not representing the ground truth and potentially even those with higher scores than the ground truth (a scenario likely to occur in settings with insufficient observational data (Friedman & Koller, 2013)), the posterior distribution may become skewed, diverging significantly from the true distribution underlying the data. In other words, it is likely that our prior over graphs matters. In DAG-GFN, the choice of a uniform prior treats all DAGs as equally likely, not incorporating any knowledge that could guide the model toward a more accurate distribution. This can lead to poor approximations of the dataset's underlying distribution, resulting in samples that may not adequately reflect the true graph.

In this paper, we introduce a new algorithm called CPDAG-GFN, which uses GFlowNet to produce sets of CPDAG candidates from observational data. Unlike DAG-GFN, which searches within the DAG space, our approach operates directly in the CPDAG space, enabling direct CPDAG sampling. Moreover, instead of relying on the posterior to produce a final set of candidates, as in DAG-GFN, we rely on the posterior as an amortized sampler from which we can select an ideal set of candidates. In particular, our approach yields relatively high top-K scoring CPDAGs from this amortized sampling. This is possible because GFlowNets can be seen as amortized samplers capable of exploring multiple high-reward states (e.g., scores) during training.

As mentioned above, relying solely on scores to prioritize graphs may lead to discrepancies with the ground truth, as high-scoring graphs might significantly deviate from it. To address this, we refine the sampled candidate graphs by incorporating additional knowledge into our CPDAG-GFN algorithm. One can think of this as imposing a prior belief into our CPDAG-GFN algorithm, though not in the Bayesian sense. Specifically, we enhance our CPDAG-GFN algorithm by applying a heuristic filter, removing the least common edges among the sampled graphs. This additional step is based on the conjecture that the top-K graphs sampled from GFlowNet often share common edge features likely present in the true graph, helping to align the top-K scoring graphs more closely with the actual CPDAG underlying the data.

---

[2] We discussed DAG-GFN in the introduction due to its use of the GFlowNet framework, which is also central to our approach, though it is not the direct motivation for our work.

[3] For further details on how DAG-GFN and CPDAG-GFN differ, refer to Appendix K.

The contributions of this paper are as follows: We introduce a novel algorithm named CPDAG-GFN, designed to learn multiple CPDAG candidates from observational data. We evaluate our method using both synthetic and real-world datasets and demonstrate that it performs competitively with established methods.

## 2 Preliminaries

In this section, we review the concepts relevant to our proposed method, CPDAG-GFN.

### 2.1 Bayesian networks, Markov Equivalence Class, CPDAGs

A **Bayesian network** (Pearl, 2009; Koller & Friedman, 2009) is a probabilistic graphical model represented by a DAG over a set of random variables $X_1, \ldots, X_n$. Each variable $X_i$ is associated with a collection of conditional distributions given its parent nodes, denoted as $\mathrm{Pa}(X_i)$ [4]. The dependency structure of a Bayesian network leverages the *Markov Property*, which asserts that each variable $X_i$ is conditionally independent of its non-descendants in the graph, given its parents $\mathrm{Pa}(X_i)$. This property allows us to factorize the joint distribution of the network into the product of conditional probabilities for each node given its parents (Jin et al., 2023):

$$P(X_1, X_2, \ldots, X_n) = \prod_{i=1}^{n} P(X_i \mid \mathrm{Pa}(X_i)) \tag{1}$$

**Markov Equivalence Classes (MECs):** DAGs that encode the same conditional independencies are Markov equivalent and are said to belong to the same MEC. These DAGs induce the same joint distribution (Jin et al., 2023). DAGs that are Markov equivalent share the same skeleton [5] and v-structures (see below for definition).

**CPDAGs:** A MEC is represented by a CPDAG, also known as an *essential graph* (Castelletti et al., 2018). A CPDAG is a type of partially directed graph that may consist entirely of directed edges, entirely of undirected edges, or a combination of both. It primarily consists of the following three types of edges:

- *Directed edge:* If an edge $x \to y$ appears in every DAG in the MEC, then the CPDAG contains a directed edge $x \to y$.
- *Undirected edge:* If the edges $x \to y$ and $y \to x$ each appear in at least one DAG in the MEC, then the CPDAG contains an undirected edge between $x$ and $y$.
- *V-structure:* If an ordered triple of nodes $(x, y, z)$ forms a configuration where $x \to y \leftarrow z$ and $x$ and $z$ are not connected by any edge, then this configuration is classified as a v-structure (Pearl, 2009).

The following theorem provides necessary and sufficient conditions for a graph to be the CPDAG of some MEC, which is essential for defining the search space for our CPDAG-GFN algorithm in Section 3.2.1.

**Theorem 1** (Andersson et al. (1997)). *A graph $G$ is a CPDAG if and only if it satisfies all of the following four conditions:*

*(a) $G$ contains no directed cycle.*
*(b) Every chain component of $G$ is chordal.*
*(c) The graph $a \to b - c$ does not occur as an induced subgraph of $G$.*
*(d) Every directed edge $a \to b$ in $G$ is strongly protected in $G$.*[6]

Note that not every partially directed graph qualifies as a CPDAG of some MEC, as indicated in the theorem above.

---

[4]$\mathrm{Pa}(X_i)$ represents the collection of values of all parent nodes; for a node without parents, this set is empty.
[5]The skeleton of a DAG refers to the undirected graph obtained by ignoring the direction of all edges in the DAG.
[6]Refer to Andersson et al. (1997) for the definition of 'strongly protected' and a review of the terms in (b) above.

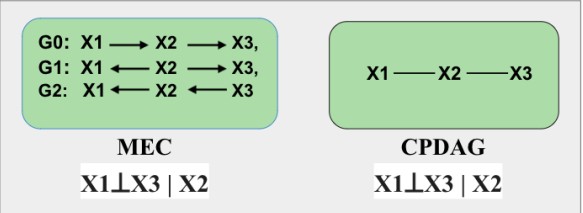

| # variables | # MECs | MECs/DAGs |
|---|---|---|
| 1 | 1 | 1.00000 |
| 2 | 2 | 0.66667 |
| 3 | 11 | 0.44000 |
| 4 | 185 | 0.34070 |
| 5 | 8782 | 0.29922 |
| 6 | 1067825 | 0.28238 |
| 7 | 312510571 | 0.27443 |
| 8 | 212133405200 | 0.27068 |
| 9 | 32626056291213 | 0.26888 |
| 10 | 111890205449597514 | 0.26799 |

Figure 1: Illustration of a MEC and its corresponding CPDAG for 3 variables. The MEC consists of multiple DAGs (i.e. $G_0$, $G_1$, $G_2$) that share identical conditional independencies, represented by $X_1 \perp X_3 \mid X_2$. The CPDAG is a single graph that compactly represents these DAGs, encapsulating the same conditional independencies.

Figure 2: The number of CPDAGs as a function of the number of variables. The column labeled 'MEC/DAG' represents the ratio between the number of CPDAGs and the number of DAGs for each variable count (Gillispie & Perlman, 2002).

## 2.2 GFlowNet

Generative flow networks (Bengio et al., 2021; 2023), or GFlowNets, were introduced as a framework to learn to sample from an unnormalized density function, typically referred to as the reward $R(s)$ in GFlowNet literature, by decomposing the generative process in a trajectory of constructive steps. GFlowNets work by modelling the *flow* that goes through the network representing the space of possible constructions, accounting for all possible construction orders of an object. We introduce the framework for a discrete setting, but continuous settings are also possible (Lahlou et al., 2023).

In a GFlowNet the state space is defined by a pointed DAG, which we denote $\mathcal{H} = (\mathcal{S}, \mathcal{A})$, with a unique initial state $s_0 \in \mathcal{S}$ and some terminal states $\mathcal{X} \subseteq \mathcal{S}$ on which $R : \mathcal{X} \to \mathbb{R}_{\geq 0}$ the reward function is defined. We define the exact state space we use in Section 3.2.1, but note that the GFlowNet DAG is distinct from the (CP)DAGs presented above; in fact, states within $\mathcal{H}$ are themselves (CP)DAGs. Elements of the action space $(s \to s') \in \mathcal{A}$ denote valid constructive steps, such as adding a directed edge in a CPDAG (see Figure 3), or may represent the action of ending generation. GFlowNets define a forward policy used for sampling objects, $P_F(s'|s)$, a backward policy $P_B(s|s')$ (a policy on the reverse Markov decision process, i.e., a model giving a distribution over the parents of any state), and an estimate of the partition function $Z$ representing the sum of all rewards–and the total flow when interpreting $\mathcal{H}$ as a network. The policy $P_F$ determines a terminating distribution $P_F^\top$ over $\mathcal{X}$, which is the marginal distribution over the final states of trajectories sampled following $P_F$ (i.e., those at which the termination action is taken).

GFlowNets are trained by driving a model to respect constraints which preserve the flow within $\mathcal{H}$. One such set of constraints are the *trajectory balance* constraints (Malkin et al., 2022), whereby for any trajectory $\tau = (s_0 \to s_1 \to ... \to s_n = x)$:

$$Z \prod_{i=1}^{n} P_F(s_{i+1}|s_i) = F(s_n) \prod_{i=1}^{n-1} P_B(s_i|s_{i+1}) \tag{2}$$

where $F(s_n) = R(x)$ is the flow of the terminal (sink) state $x$. With the above constraints satisfied, sampling transitions starting from $s_0$ and using $P_F$ guarantees that the marginal terminating distribution $P_F^\top(x) \propto R(x)$.

In the present work we parameterize $\log Z_\theta$ and $P_F(\cdot|\cdot; \theta)$, use a uniform $P_B$, and apply the trajectory balance objective (Malkin et al., 2022), which for a trajectory $\tau = (s_0 \to s_1 \to ... \to s_n = x)$ is:

$$\mathcal{L}_{\text{TB}}(\tau) = \left( \log \frac{Z_\theta \prod_{i=0}^{n-1} P_F(s_{i+1}|s_i; \theta)}{R(x) \prod_{i=0}^{n-1} P_B(s_i|s_{i+1})} \right)^2. \tag{3}$$

When training with GFlowNet objectives, one samples trajectories from some behaviour policy (which may either coincide with $P_F$ – *on-policy training*, as done in this paper – or use off-policy exploration) and performs gradient descent steps on the loss, in our case the one in (3).

## 3 Method

The goal of CPDAG-GFN is to return multiple CPDAGs that approximate the true CPDAG underlying the data. The objective of using GFlowNet is to construct a posterior distribution over CPDAGs that will allow us to sample K high-reward CPDAGs. GFlowNet is suitable for this because of its capabilities as an amortized sampler, which allows for sampling during the training process and facilitates the exploration of high-reward CPDAGs throughout. Since the top K sampled high-scoring CPDAGs may often not align well with the true CPDAG underlying the data Koller & Friedman (2009), we incorporate a heuristic filter into our algorithm.

### 3.1 Heuristic Edge-Sparsity Filter

We introduce a heuristic filter that removes the $L$ least common edges among the top K sampled graphs at the end of training, while ensuring that each removal does not violate CPDAG properties in 1, thereby maintaining the graph's validity as a CPDAG. $L$ is a hyperparameter. This approach is motivated by our observation that edges consistently appearing across top K high-scoring models tend to reflect shared patterns in the true underlying graph. This observation led us to hypothesize that high-scoring graphs may share common edge features with the true CPDAG. The filter targets edges that do not consistently appear across models, which may mitigate the presence of spurious edges in the sampled graphs.

### 3.2 GFlowNet setup

The setup for GFlowNet includes defining a state space, a reward function, a graph neural network (GNN), and a loss function. For the loss function, we employ the trajectory balance function, which is covered in Section 2.2. We will now discuss each of these components in turn.

#### 3.2.1 State space

We define the state space to consist solely of CPDAGs, and the GFlowNet's action space as transitions from one CPDAG graph to the next.

Recall from Section 2.1 that a CPDAG may consist entirely of undirected edges, solely of directed edges, or a combination of both directed and undirected edges. We thus use the following three actions: add a directed edge, add an undirected edge, apply the `makeV` operator which transforms a graph structure from $x - y - z$ to $x \to y \leftarrow z$. In Appendix A, we show that these three actions are enough to construct any CPDAG in the search space.

In CPDAG-GFN, graphs are built starting from an edge-less graph $s_0$, with transitions to a new state achieved by applying one of the three actions mentioned above. For any given state, we limit allowable actions to those that lead to a new graph satisfying all the CPDAG properties outlined in Theorem 1 section 1. This ensures that the graph resulting from any permitted action will also be a CPDAG.

Additionally, we introduce a stop action, which serves as the termination point for a trajectory through the state space. If a stop action is sampled in $s_i$, we consider the state terminal and compute its reward $R(s_i)$ (see Figure 3).

#### 3.2.2 Reward function

Let $D$ represent a dataset of $N$ *i.i.d.* observations. Since we aim to sample CPDAGs that fit the data well, we define the reward as the score function $R(G) = score(G, D)$. We follow the definition of the score function by Koller & Friedman (2009):

$$score(G, D) = P(D \mid G)P(G), \tag{4}$$

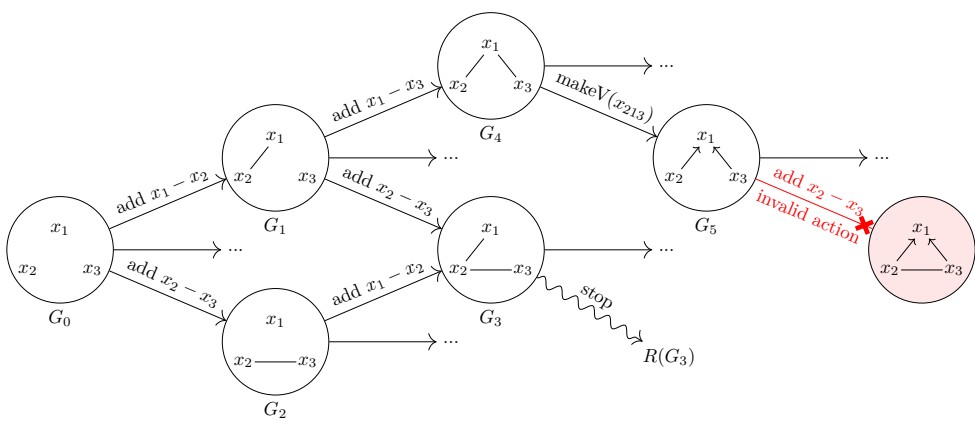

Figure 3: This figure illustrates the CPDAG construction process during GFlowNet training. It shows transitions from an initial edge-less graph state $G_0$ to subsequent states, distinguishing between valid actions that lead to new CPDAGs and invalid actions prohibited because they result in non-CPDAG states. Sampling a *stop* action at state $G_3$ concludes the trajectory, with the reward $R(G_3)$ then computed.

where $P(G)$ is a structure prior which we set to be uniform. This configuration enables GFlowNet to explore the space of CPDAGs without any initial bias, and is a common choice (Eggeling et al., 2019; Koller & Friedman, 2009; Deleu et al., 2022).

The marginal likelihood $P(D \mid G)$ can be calculated using any score-equivalent function such as the BGe score (Kuipers et al., 2014; Geiger & Heckerman, 1994), the BDe score (Heckerman et al., 1995; Chickering, 2013).[7] To obtain a score for a CPDAG, we first find a DAG belonging in the MEC that this CPDAG represents using an algorithm by Dor & Tarsi (1992), then assign that score to the CPDAG.

### 3.2.3 Parameterization with graph neural networks

In GFlowNet, we need to learn a policy that gives the probability of each state. Considering the exponential number of CPDAG states, it is impossible to store all these probabilities in a lookup table. Meanwhile, the training trajectories may not cover every state in the CPDAG space, which means we have to predict the probability for unseen states at test time. Following previous works Bengio et al. (2021); Deleu et al. (2022), we parameterize the policy with a neural network over the graph structure of the current state.

Considering the directed nature of CPDAGs, we employ a relational graph neural network (RGCN) Schlichtkrull et al. (2018) to encode the node representations, where directed and undirected edges are treated as different relations. For any node $u$, RGCN iteratively computes its representation $\boldsymbol{h}_u$ with the following message passing step

$$\boldsymbol{h}_u^{(t+1)} = \sigma \left( \sum_{r \in \mathcal{R}} \sum_{v \in \mathcal{N}_r(u)} \frac{1}{|\mathcal{N}_r(u)|} \boldsymbol{W}_r^{(t)} \boldsymbol{h}_v^{(t)} + \boldsymbol{W}_0^{(t)} \boldsymbol{h}_u^{(t)} \right) \tag{5}$$

where $\mathcal{R} = \{\text{directed}, \text{directed}^{-1}, \text{undirected}\}$ is the set of relations, $\mathcal{N}_r(u)$ is the set of nodes connected by relation $r$ from node $u$, $\boldsymbol{W}_r^{(t)}$ and $\boldsymbol{W}_0^{(t)}$ are learnable matrices and $\sigma$ is the activation function. The input embeddings $\boldsymbol{h}_u^{(0)}$ are initialized with one-hot embeddings to distinguish different nodes in the graph.

The three actions in Sec.3.2.1 correspond to link prediction and graph classification tasks on graph structure. Therefore, we decode the actions of adding edges with SimplE score function Kazemi & Poole (2018), a common choice for relational link prediction. Specifically, SimplE computes the following for a head node $u$,

---

[7]Other score-equivalent functions – that is, those that assign the same score for any two Markov-equivalent graphs such as the Akaike information criterion (AIC) (Akaike, 1974), the Bayesian information criterion (BIC) (Schwarz, 1978), the minimum description length (MDL) (Rissanen, 1986), etc.

a relation $r$ and a tail node $v$

$$score(u, r, v) = (\boldsymbol{h}_u \odot \boldsymbol{r}_r)^\top \boldsymbol{K} \boldsymbol{h}_v \tag{6}$$

where $\boldsymbol{r}_r$ is a learnable embedding for relation $r$, $\odot$ is element-wise multiplication and $\boldsymbol{K}$ is an anti-diagonal identity matrix. Thanks to the asymmetry of the anti-diagonal kernel, SimplE can have different predictions for $(u, r, v)$ and $(v, r, u)$, thereby is suitable for the action of adding directed edges in our model. To accommodate the action of adding a v-structure, we consider it as a link prediction problem between a set of two nodes $u_1$, $u_2$ and a collider node $v$, i.e. predicting $(\{u_1, u_2\}, r, v)$. The stop action is modeled by a multi-layer perceptron (MLP) over the graph representation, which is obtained by a min pooling operation applied over all node representations in the graph.

## 4 Experiments

In this section, we empirically assess CPDAG-GFN's performance against established methods by comparing the learned CPDAGs to a real-world proteomics dataset obtained from protein signaling networks (Sachs et al., 2005). Additionally, we evaluate our method on synthetic datasets across different settings (e.g., data size, noise level, network topology). It is important to note that our goal is not to analyze the specific conditions under which our method outperforms baselines, but rather to evaluate CPDAG-GFN's performance in different scenarios. Experimental results show that CPDAG-GFN performs competitively against established baselines in these settings.

### 4.1 Experimental evaluation

**Evaluation metric** We adopt the evaluation metrics used in prior work (Deleu et al., 2022; Lorch et al., 2021). We assess the performance of each algorithm using the Expected Structural Hamming Distance (E-SHD) — which measures the discrepancy between the inferred and the true CPDAGs (see Appendix B for further detail), with lower E-SHD indicating better performance—and the Area Under the Receiver Operating Characteristic curve (AUROC), where a higher value signifies better performance. Additionally, we compute the average Structural Hamming Distance (SHD) between all pairs of CPDAGs in the sample, providing a measure of average dissimilarity across the CPDAGs. While this distance is not a formal evaluation metric, it provides insight into the diversity of the sampled graphs, with a higher average indicating greater dissimilarity.

**Baselines** To provide a comprehensive evaluation of our proposed method, we have selected baselines that use different approaches for approximating distributions over structural models. We include bootstrapping with PC (Spirtes et al., 2001) and Greedy Equivalence Search (GES) (Chickering, 2002b). Additionally, we employ DAG-GFN (Deleu et al., 2022), which leverages GFlowNet; DiBS (Lorch et al., 2021), which uses a variational inference approach; and MC3 (Madigan et al., 1995; Giudici & Castelo, 2003), representing MCMC-based methods.

### 4.2 Evaluation on synthetic data

In our experiments, we use the BGe score function as our reward function. Following Lorch et al. (2021), we generate ground truth networks using linear-Gaussian Bayesian networks, with their structures sampled according to the Erdős-Rényi (ER) model (ERDdS & R&wi, 1959) and the scale-free (SF) model (Barabási & Albert, 1999). These models were selected for their contrasting structural properties to ensure a diverse evaluation of CPDAG-GFN's adaptability and efficacy across different network topologies. In addition, we designed our experiments to include several scenarios: different observational data sizes ranging from small (100) to large (a million) to demonstrate scalability, varying levels of noise in the data from small (0.01) to moderate (0.1), and different network complexities with expected degrees of $1d$, $2d$, and $3d$, where $d$ is the number of nodes, in which we set to d=10 in our experiments. For each experimental run, we sample K graphs, with K set to 100. Performance metrics (e.g. E-SHD and AUROC) are derived from 10 distinct datasets, each one generated from a unique Bayesian network. Performance results are presented in Figures 1–5, which indicate that our algorithm performs competitively.

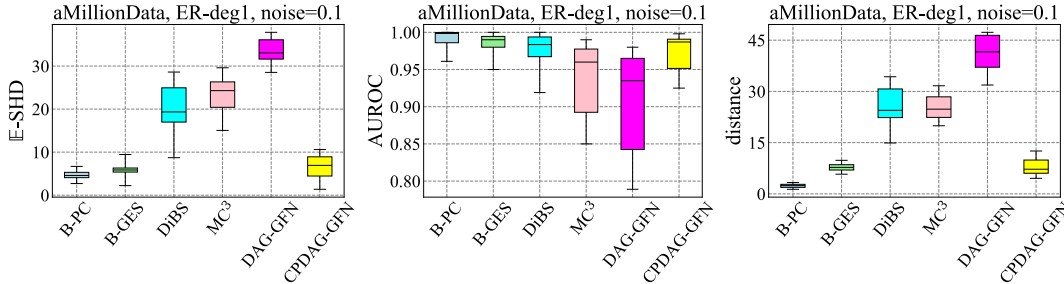

Figure 4: Comparison of E-SHD and AUROC metrics for a dataset generated from 1 million observations using a ground truth graph sampled from an Erdős-Rényi model (ER-deg1) with a noise level of 0.1. Lower E-SHD and higher AUROC are preferred. A higher distance in the third figure indicates greater dissimilarity among the graphs.

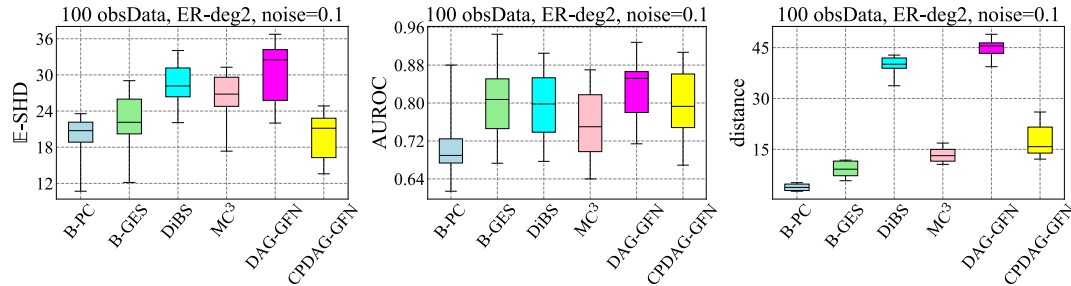

Figure 5: Comparison of E-SHD and AUROC metrics for a dataset generated from 100 observations using a ground truth graph sampled from an Erdős-Rényi model (ER-deg2) with a noise level of 0.1. Lower E-SHD and higher AUROC are preferred. A higher distance in the third figure indicates greater dissimilarity (e.g. more diversity) among the graphs.

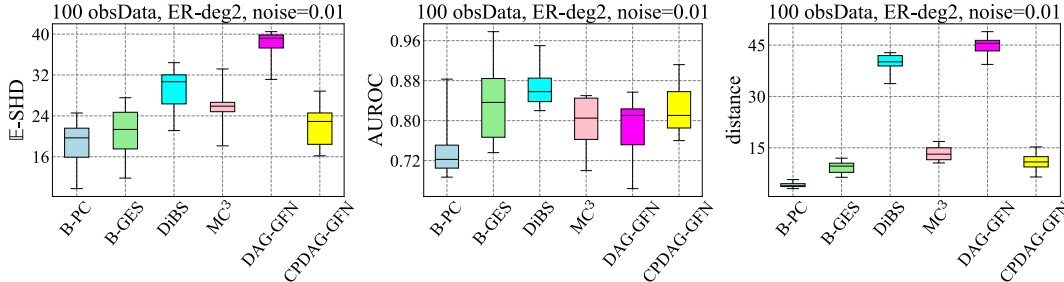

Figure 6: Comparison of E-SHD and AUROC metrics for a dataset generated from 100 observations using a ground truth graph sampled from an Erdős-Rényi model (ER-deg2) with a noise level of 0.01. Lower E-SHD and higher AUROC are preferred. A higher distance in the third figure indicates greater dissimilarity among the graphs.

### 4.3 Real world data: Protein network from cell data

A widely used benchmark in structure learning is the Sachs dataset (Sachs et al., 2005), together with its reference network of 11 proteins and 17 edges. Sachs et al. (2005) used Bayesian network inference with intervention data to reconstruct signaling pathways and compared their learned network to a reference network assembled from prior biological pathways reported in the literature. Their reconstruction largely reflects these pathways, although three well-established edges (PIP$_2$ → PKC, PLC-$\gamma$ → PKC, PIP$_3$ → Akt) were not recovered. This reference network, together with the Sachs dataset, has since become a widely

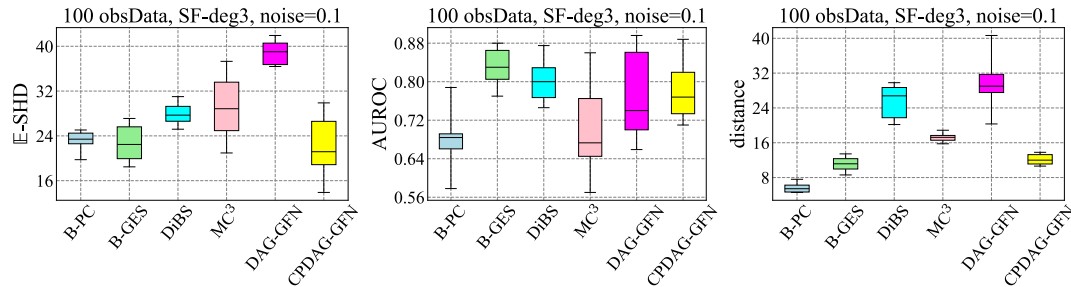

Figure 7: Comparison of E-SHD and AUROC metrics for a dataset generated from 100 observations using a ground truth graph sampled from a scale-free model (SF-deg3) with a noise level of 0.1. Lower E-SHD and higher AUROC are preferred. A higher distance in the third figure indicates greater diversity among the graphs.

| Method | $\mathbb{E}[SHD]$ | AUROC | $\mathbb{E}[\#\ Edges]$ |
|---|---|---|---|
| MC$^3$ | $39.32 \pm 1.79$ | $0.645 \pm 0.041$ | $39.46 \pm 0.97$ |
| Bootstrap GES | $19.74 \pm 0.097$ | $0.751 \pm 0.011$ | $11.11 \pm 0.090$ |
| Bootstrap PC | $17.61 \pm 0.23$ | $0.728 \pm 0.019$ | $8.45 \pm 0.335$ |
| DiBS | $\mathbf{13.28 \pm 0.17}$ | $0.756 \pm 0.017$ | $11.47 \pm 0.34$ |
| DAG-GFN | $18.92 \pm 0.019$ | $0.658 \pm 0.019$ | $23.02 \pm 0.14$ |
| CPDAG-GFN | $16.61 \pm 0.65$ | $\mathbf{0.757 \pm 0.029}$ | $\mathbf{15.42 \pm 0.61}$ |

Table 1: Inference of protein signaling pathways from cytometry data  (Sachs et al., 2005) Metrics are the mean $\pm$ SD of 10 experimental run

used benchmark in the causal discovery community, providing a common basis for evaluating algorithms on real-world biological data.

In this section, we evaluate our work using this widely used benchmark. The dataset consists of $N = 854$ continuous observations. The results, presented in Table 1, compare the evaluation metrics E-SHD and AUROC of CPDAG-GFN against the baselines. For AUROC, a higher value is better, and for E-SHD, a lower value is better.

CPDAG-GFN demonstrates competitive performance against the other methods, achieving an E-SHD of $16.61 \pm 0.65$. Notably, while DiBS achieves the lowest E-SHD, it predicts only 11.47 edges on average. In contrast, CPDAG-GFN's edge count of $15.42 \pm 0.81$ is closest to the ground truth network, which consists of 17 edges. Although AUROC values for DiBS, Bootstrap GES, and CPDAG-GFN are relatively close, CPDAG-GFN performs competitively by achieving a relatively low E-SHD, high AUROC, and an edge count closer to the ground truth compared to other baselines.

## 5    Related work

**Markov Chain Monte Carlo (MCMC)**   Earlier papers that explored structure learning in the space of CPDAGs by means of MCMC include: Madigan et al. (1996) and Castelo & Perlman (2004). A more recent approach is by Castelletti et al. (2018). The paper focuses on learning sparse CPDAGs. To enhance the structure learning of these CPDAGs and ensure the graphs remain sparse, the authors introduce a sparsity constraint. This constraint limits the CPDAG space to a subspace where the number of edges does not exceed 1.5 times the number of nodes in the graph. While this approach can be effective, it can be a drawback in scenarios where the data-generating process is more complex and less sparse. In such cases, the constraint may bias the learning towards sparse graph that fail to capture more complex relationships within the data. Another draw back of using MCMC methods is their tendency to become trapped in a single mode of high probability  (Syed, 2022), restricting their ability to explore diverse graph structures across different regions

of high-probability modes. By using the GFlowNet approach instead of MCMC, the method we propose mitigates being confined to a single mode.

**Point estimate methods** Two widely known point estimate methods for computing a CPDAG from observational data are constraint-based methods and score-based methods. Constraint-based methods, such as the PC (Peter and Clark) algorithm and the Fast Causal Inference (FCI) algorithm (Spirtes et al., 2001), rely on conditional independence tests to identify the CPDAG that represents causal structures consistent with a given dataset (Eberhardt, 2017). In contrast, score-based methods, such as the Greedy Equivalence Search (GES; Chickering (2002b)), rely on a score function. These methods traverse the space of CPDAGs, assigning scores to graphs to measure their fit to the data. At each step, an edge is either added, removed, or reversed. As the name suggests, the search is greedy, choosing the state with the highest score to progress.

**Amortized sampling methods** Recent approaches to Bayesian structure learning that approximate the posterior distribution over DAGs include DAG-GFN (Deleu et al., 2022) and DiBS (Lorch et al., 2021), both operate within the space of DAGs. The DiBS method employs a variational approach, representing DAGs in a continuous latent space to facilitate the learning of the posterior distribution over network structures. Conversely, the DAG-GFN method uses GFlowNet to achieve a posterior distribution over DAGs where probabilities are approximately proportional to assigned rewards. CPDAGs can be obtained by converting the sampled DAGs to CPDAGs.

## 6 Conclusion

We have introduced a novel method for learning CPDAG candidates underlying observational data using GFlowNet. To better align the candidates with the ground truth, we applied a heuristic filter by removing the least common edges from the sample. Unlike traditional methods that produce a single CPDAG, our approach generates multiple CPDAG candidate structures by sampling directly from CPDAG distributions. In future work, we aim to explore the integration of domain-specific knowledge into the CPDAG-GFN algorithm. This could involve incorporating Bayesian priors to guide the sampling process toward more plausible causal structures, or developing heuristic-based constraints that reflect expert knowledge. By further incorporating specific causal hypotheses within the model, we hope to improve the accuracy and relevance of the generated CPDAGs, particularly in applications where expert knowledge is available.

## Acknowledgments

We would like to thank Kolya Malkin for the fruitful discussions that led to the proof in Appendix A and for his feedback on our manuscript. We also thank members of the Mila community, including Mizu Nishikawa-Toomey, Mansi Rankawat, Tristan Deleu, Fabrice Normandin, Jithendaraa Subramanian, Anirudh Goyal, Guillaume Huguet, Paul Bertin, Alex Tong, Jason Hartford, Simon Lacoste-Julien, Antonio Gois, Ionelia Buzatu, Philippe Brouillard, Michael Przystupa, Victor Schmidt, Shruti Joshi, Divyat Mahajan, and Rabiul Awal for their help and support.

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

## A    Sufficiency of actions to traverse CPDAG search spaces

In this section, we demonstrate that the three traversal actions—adding a directed edge, adding an undirected edge, and applying 'make V' operators—are sufficient to reach any CPDAG within the search space. To do so, we introduce the following two propositions:

**Proposition 2** (Andersson et al. (1997))**.** *Let $G$ and $H$ be two CPDAGs graphs with the same vertex set $V$. Then there exists a finite sequence $G \equiv G_1, \dots, G_k \equiv H$ of CPDAGs with vertex set $V$ such that each consecutive pair of $G_i, G_{i+1}$ differs by one of the three traversal traversal actions: add directed edge, add undirected edge, or makeV.*

Building upon the aforementioned proposition, we present the following proposition:

**Proposition 3.** *For any essential $G$, there exists a sequence of graphs $G_0, \dots, G_n$ such that:*

- *$G_0$ is the graph with no edges, and $G_n = G$.*
- *Each $G_i$ is a CPDAG.*
- *For each $i$, $G_{i+1}$ can be obtained from $G_i$ by one of the operators: add directed edge, add undirected edge, or makeV.*

*Proof.* This is immediate from the construction in the proof of Andersson's Proposition 4.5, which provides an algorithm for constructing such a sequence for any $G$. Note that the statement of Andersson's Proposition 4.5 alone does not imply this result, as it only guarantees a sequence $G_i$ such that for each $i$, either $G_i$ is obtained from $G_{i+1}$ by one of the operators, or $G_{i+1}$ is obtained from $G_i$ by one of the operators. $\qquad\square$

Proposition 3 above asserts that we can traverse the entire search space of CPDAGs starting from the empty graph $G_0$ and using only the three traversal actions (add a directed edge, add an undirected edge, or makeV operation), i.e., every state is reachable from the initial state. Each traversal action either increases the number of edges or does not change the number of edges while increasing the number of directed edges, which implies the search space is acyclic.

## B    Evaluation Metrics

We adopt the evaluation metrics used in prior work (Deleu et al., 2022; Lorch et al., 2021): the Expected Structural Hamming Distance (E-SHD) to the ground truth graph $\mathcal{G}^*$, and the Area Under the Receiver Operating Characteristic Curve (AUROC). Following prior work (Deleu et al., 2022; Lorch et al., 2021) the AUROC is computed by thresholding the posterior edge $p(g_{ij} \mid D)$ at varying thresholds.

Similarly, the E-SHD to the ground truth graph $(\mathcal{G}^*)$ is defined as:

$$\text{E-SHD} \approx \frac{1}{n} \sum_{k=1}^{n} \text{SHD}(\mathcal{G}_k, \mathcal{G}^*)$$

where $n$ represents the number of unique CPDAGs sampled from the posterior, and $\mathcal{G}_k$ denotes a CPDAG. The $\text{SHD}(\mathcal{G}_k, \mathcal{G}^*)$ counts the number of edge changes that separate the learned CPDAGs $\mathcal{G}_k$ from the ground truth $\mathcal{G}^*$ (Peters & Bühlmann, 2015; Lorch et al., 2021).

## C  Additional experiments

The ground-truth graphs are sampled according to an Erdos-Rényi model with average degrees equal to 1, 2, and 3, respectively, denoted ER-deg1, ER-deg2, and ER-deg3. Each experiments is ran between 7 to 9 seeds. The DAG-GFN was run using the publicly available code from Deleu et al. (2022).

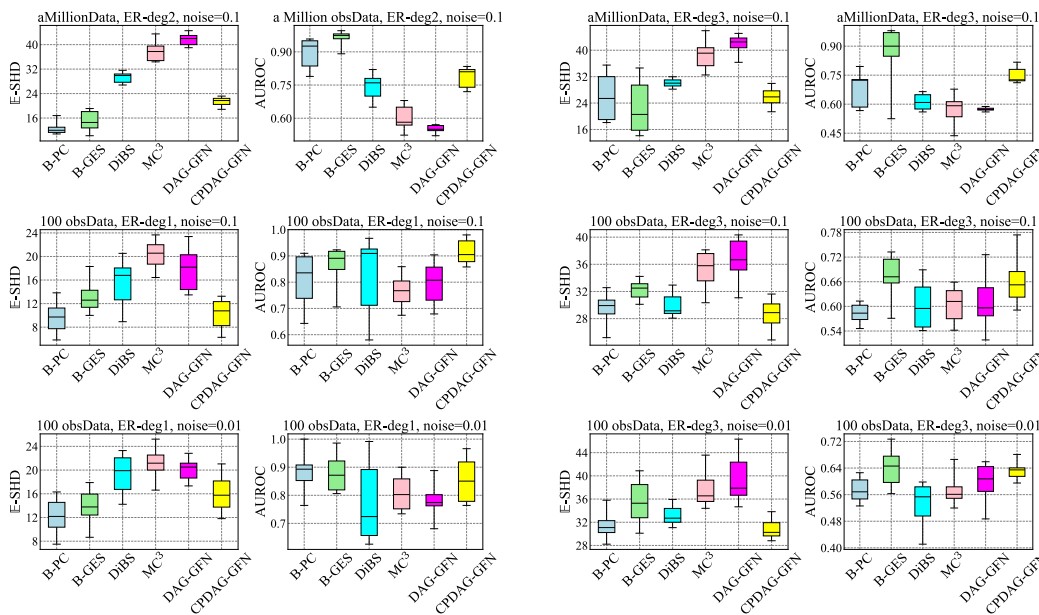

For the following experiments, the ground-truth graphs are sampled according to an scale-free model with average degrees equal to 1 and 2 respectively, denoted SF-deg1 and SF-deg2.

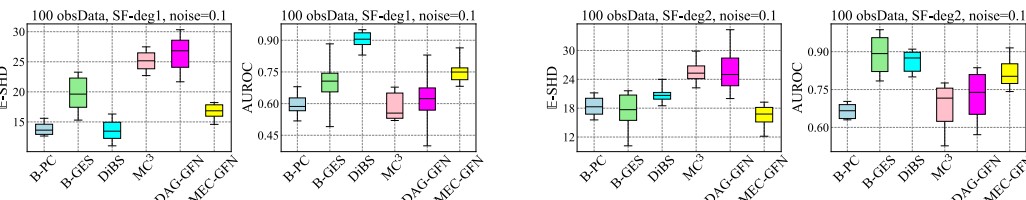

# D  DAG vs CPDAG comparison

To ensure a fair comparison, both DAG-GFN and CPDAG-GFN were implemented with the same setup, including the same optimizer, neural network architecture, and dataset. We used the trajectory balance (TB) loss and BGe score function to train both models. Comparison of E-SHD and AUROC metrics for a dataset generated from large (e.g. 1 million) and small (e.g. 100) observations using a ground truth DAG sampled from an Erdős-Rényi model (ER-deg1) and scale-free model.

Since there are two important differences between Deleu et al. (2022)'s DAG-GFN and our work, namely the top-K sampling evaluation and the heuristic filtering, we compare four settings

1. Top-$K$ DAGs obtained using Gflownet as an amortized sampler in the DAG space (call it topK-DAG) versus top-$K$ CPDAGs obtained using GFlownet as an amortized sampler in the CPDAG space (topK-cpdag).

2. rand-DAG obtained by randomly sampling K DAGs from the posterior distribution learned by DAG-GFN (Deleu et al., 2022), versus rand-CPDAG obtained by directly random sampling K cpdags from the CPDAG posterior at the end of training.

3. The same as in 1, but with the heuristic filter applied.

4. The same as in 2, but with the heuristic filter applied.

Baselines with a plus sign in front indicate heuristic filter has been applied.

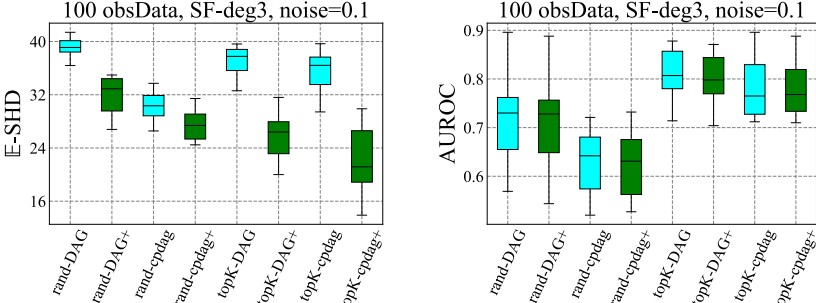

Figure 8: d=10 variables. Lower E-SHD and higher AUROC indicate better performance compared to those with higher E-SHD and lower AUROC.

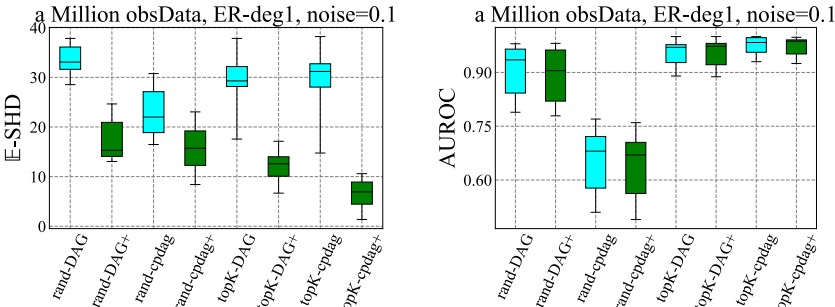

Figure 9: d=10 variables. Lower E-SHD and higher AUROC indicate better performance compared to those with higher E-SHD and lower AUROC.

# E   Performance Metrics Across Varying L Values

In this section, we analyze how CPDAG-GFN's performance (e.g. E-SHD, AUROC, and distance) varies with different values of L.

**Experimental setup for hyperameter L anaysis:** In the plots below, a unique Bayesian network was generated using different random seeds to create the data. This data was then used as input to our CPDAG-GFN algorithm to learn 100 CPDAG candidates. We computed E-SHD, AUROC, and distance metrics using these 100 CPDAGs for varying values of L. The results are presented in the plots below. All plots were generated using data from 1 million observations, ER-deg1, noise=0.1, and 10 variable setting.

**Observation:** The plots start at $L = 0$ (e.g., no least common edge is removed). As the hyperparameter $L$ increases, E-SHD initially decreases but begins to rise beyond a certain point (around $L = 40$). Similarly, AUROC starts off high but begins to drop as $L$ exceeds a specific threshold. On the other hand, the distance plot demonstrates an overall decreasing trend, with a noticeable bump in the purple scatter plot between $L = 30$ and $L = 40$. This suggests that removing certain edges may have increased the diversity of the graphs in the sample. Despite this bump, the general trend is downward, as expected, since removing the least common edges typically leads to greater similarity among the graphs, resulting in a decrease in the distance metric.

A suitable range for $L$ would be one that gives low E-SHD and high AUROC, and sufficient diversity in the generated graphs. Having said that, $L$ values ranging between 30 and 40 would be a reasonable choice.

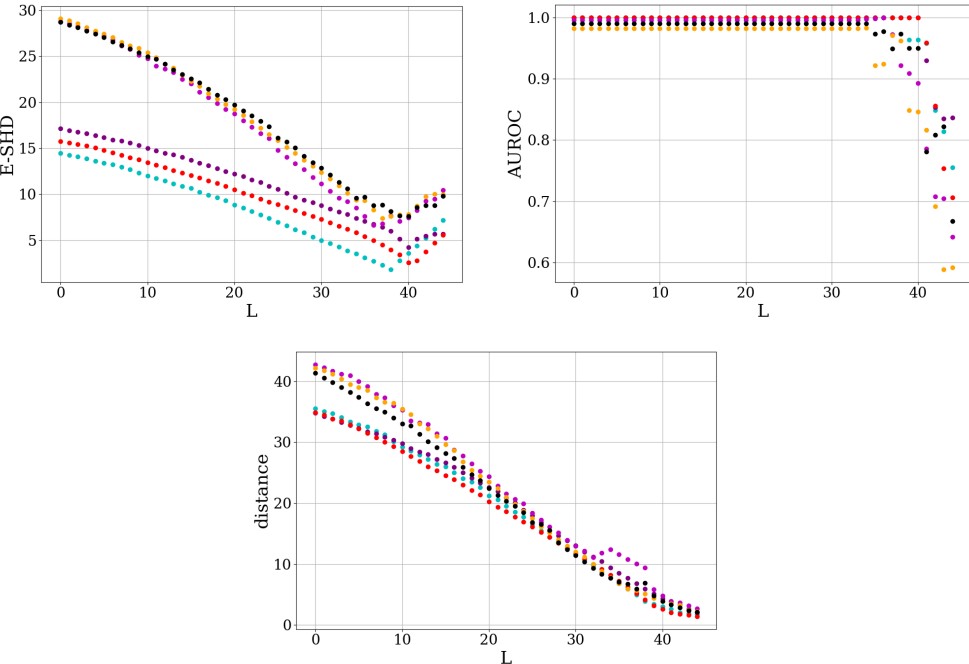

Figure 10: Lower E-SHD and higher AUROC indicate better performance.

## F    Additional experiments on heuristic filter on other baselines

While the heuristic filter was originally designed as an integral part of CPDAG-GFN, we conducted an analysis to assess whether applying the same refinement step to other baseline methods would generally lead to performance improvements in different settings. In this section, we evaluated baseline performance before and after applying the filter and reported the observed effects. Our results in different settings show that the the heuristic filter improves E-SHD across all baselines by lowering it. The largest improvements in E-SHD are observed in CPDAG-GFN and Top-K DAGs [8]. The results below highlight that while the heuristic filter provides general improvements across baselines, its effects are pronounced when combined with GFlowNet's amortized sampling approach.

The figures below compare baseline performance before (green box) and after applying the heuristic filter (blue box, indicated by a plus sign).

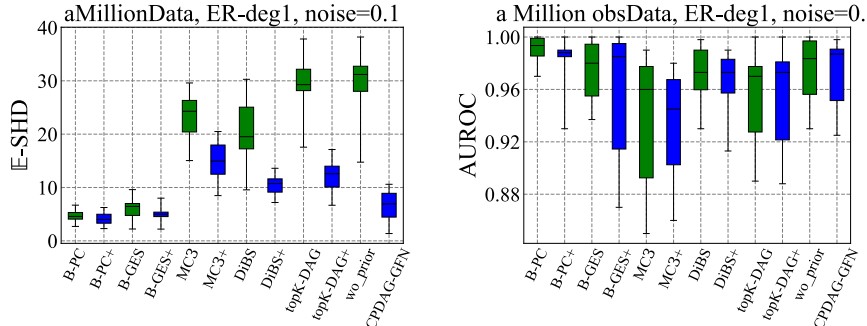

Figure 11: Baselines with lower E-SHD and higher AUROC indicate better performance compared to those with higher E-SHD and lower AUROC. Baselines with a plus sign (e.g. blue box) indicate the application of heurisitc filer, and green box is without.

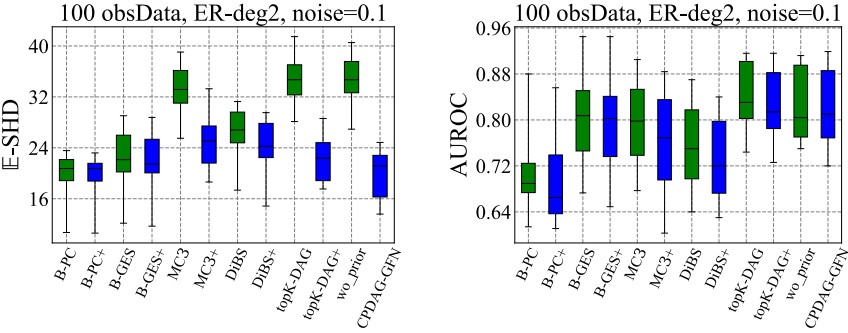

Figure 12: Baselines with lower E-SHD and higher AUROC indicate better performance compared to those with higher E-SHD and lower AUROC. Baselines with a plus sign (e.g. blue box) indicate the application of heurisitc filer, and green box is without.

---

[8]Top-$K$ DAGs are obtained using Gflownet as an amortized sampler to sample top K high scoring graphs in the DAG space and then converted to CPDAGs. It's not to be mistaken for DAG-GFN by Deleu et al. (2022).Moreover topK-DAG+ is the topK-DAG with the heuristic filter applied

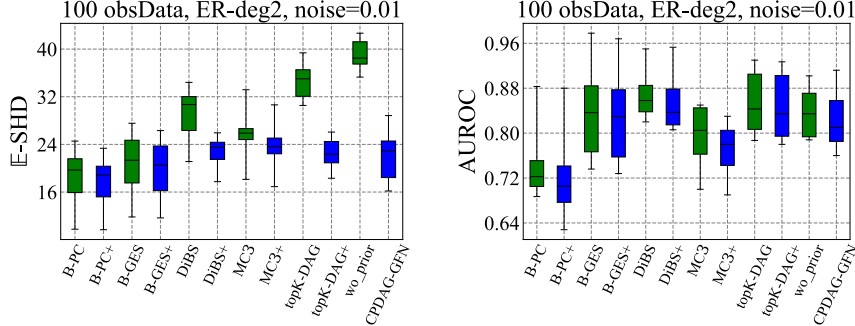

Figure 13: Lower E-SHD and higher AUROC indicate better performance compared to those with higher E-SHD and lower AUROC.Baselines with a plus sign indicate the application of the least common edge removal technique, and wo-prior represents CPDAG-GFN without the heuristic filter.

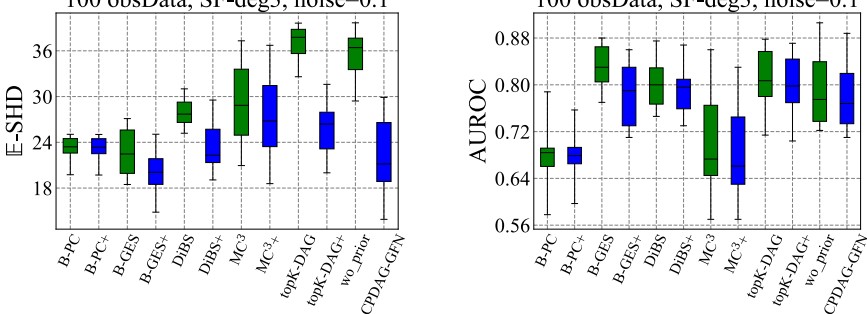

Figure 14: Baselines with lower E-SHD and higher AUROC indicate better performance compared to those with higher E-SHD and lower AUROC. Baselines with a plus sign (e.g. blue box) indicate the application of heurisitc filer, and green box is without.

# G   Defining the Mask Over Actions with CPDAG constraints

In this section, we discuss how we enforce theorem 1 of the CPDAG at each state.

To enforce the directed cycle constraint, we leverage the adjacency matrix, transitive closure, and outer product as inspired by Deleu et al. (2022) Appendix C. This process generates a mask matrix, $\text{Mask}_{\text{matrix}}$, with entries of 0 or 1. We adopt the convention that entry (row, col) in the $\text{Mask}_{\text{matrix}}$ equal to 0 indicates that the directed edge row $\rightarrow$ col will not create a directed cycle, while an entry of 1 means it will.

We identify the indices of all zero entries (a,b) in $\text{Mask}_{\text{matrix}}$. Each of these indices is fed into the flow chart[9], which carries out the remaining constraints in theorem 1. The flowchart outputs 0 (indicating the action is allowed) or 1 (indicating the action is forbidden), and the $\text{Mask}_{\text{actions}}$ array is updated accordingly.

The purpose of $\text{Mask}_{\text{actions}}$ is to filter out all invalid actions so that only valid actions are sampled. Once a valid action is sampled and applied to generate a new CPDAG, the adjacency matrix, transitive closure, and mask matrix are updated to reflect the new state. Consequently, $\text{Mask}_{\text{actions}}$ is also updated accordingly.

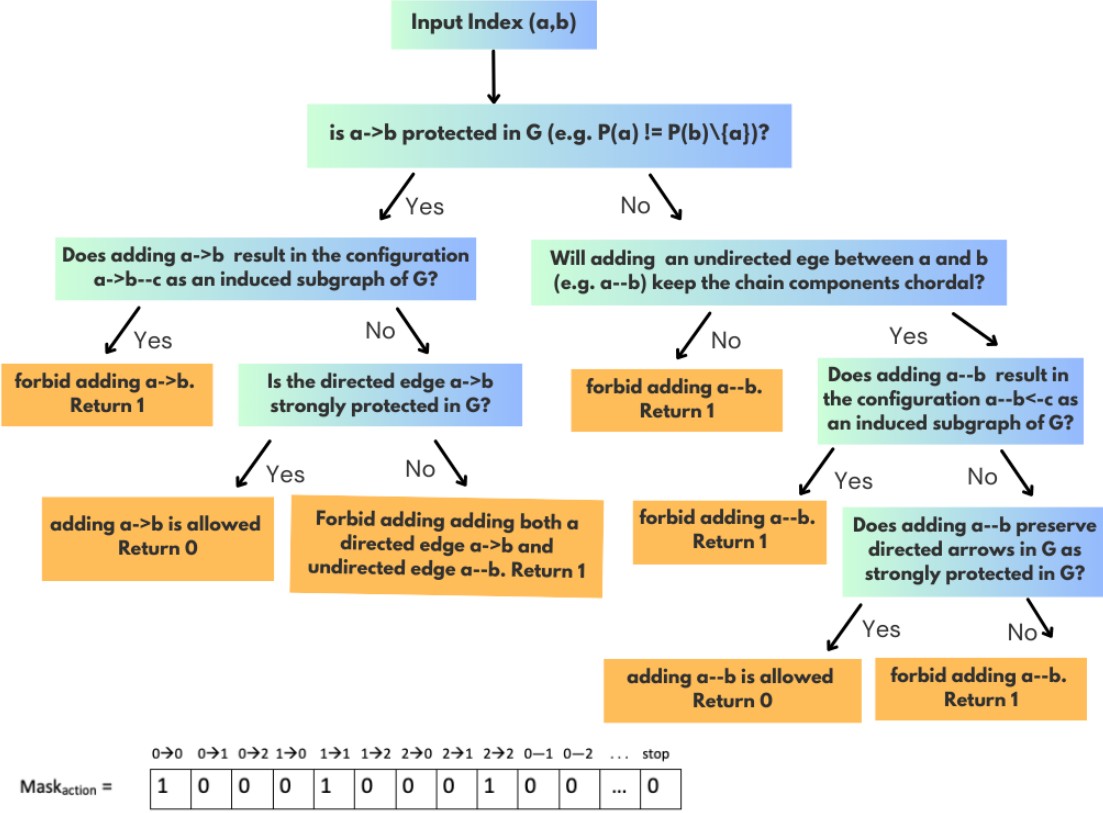

Figure 15: a) Flowchart illustrating how actions are validated and filtered at each state using the constraints outlined in Theorem 1. The input index(a,b) are the zero entries in the $\text{Mask}_{\text{matrix}}$. Here, $a \rightarrow b$ denotes a directed edge between node a and b in graph G, while $a - -b$ denotes an undirected edge between them. b) The $\text{Mask}_{\text{actions}}$ array entries are updated based on the flowchart's output, with 0 indicating a valid action and 1 indicating an invalid action.

---

[9]Please refer to Andersson et al. (1997) for further foundational and implementation details.

# H   Additional baseline: NOTEARS and MMHC

In this section, we compare our work to two well-known point estimate methods that return a single graph: DAGs with NO TEARS (NOTEARS) (Zheng et al., 2018), and the Max-Min Hill-Climbing (MMHC) Bayesian Network Structure Learning Algorithm (Tsamardinos et al., 2006).

NOTEARS is a differentiable DAG learning method that formulates the structure learning problem as a continuous constrained optimization task. Instead of searching over the combinatorial space of DAGs, it introduces a smooth and exact characterization of acyclicity using a matrix function constraint. This allows gradient-based optimization to be applied directly to learn the DAG structure. MMHC, on the other hand, is a hybrid method that combines constraint-based and score-based approaches. It first identifies the skeleton of the graph using the Max-Min Parents and Children (MMPC) algorithm, a constraint-based method. It then applies a greedy hill-climbing search with a Bayesian scoring function to orient the edges. We refer readers to the references for further details of the methods.

Both NOTEARS and MMHC are point estimate methods that return a single graph rather than a distribution over possible structures. Consequently, the E-SHD metric reduces to standard SHD in this case, and the distance metric (e.g., the average SHD between sampled graphs) is not applicable.

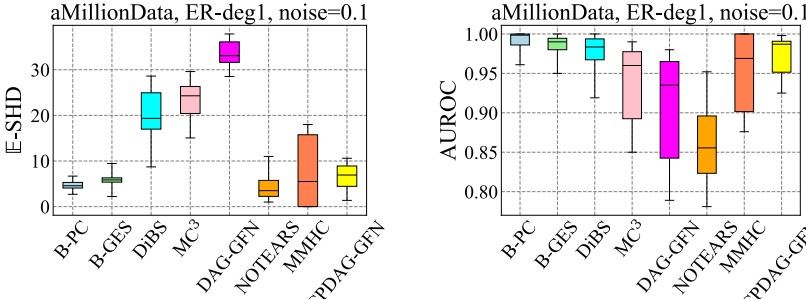

Figure 16: Lower E-SHD and higher AUROC indicate better performance compared to those with higher E-SHD and lower AUROC.

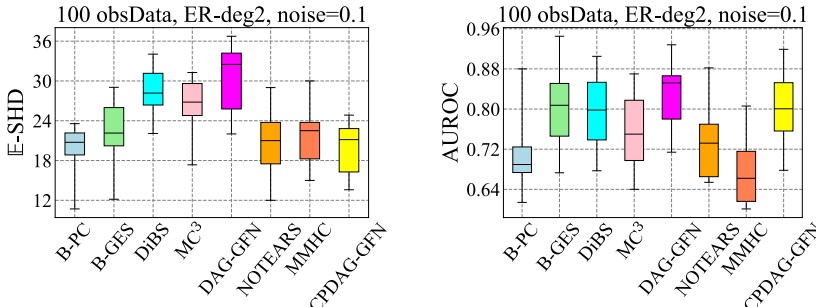

Figure 17: Lower E-SHD and higher AUROC indicate better performance compared to those with higher E-SHD and lower AUROC.

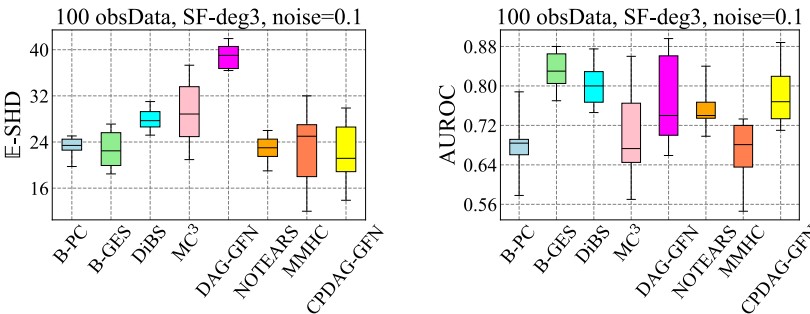

Figure 18: Lower E-SHD and higher AUROC indicate better performance compared to those with higher E-SHD and lower AUROC.

# I ROC curve

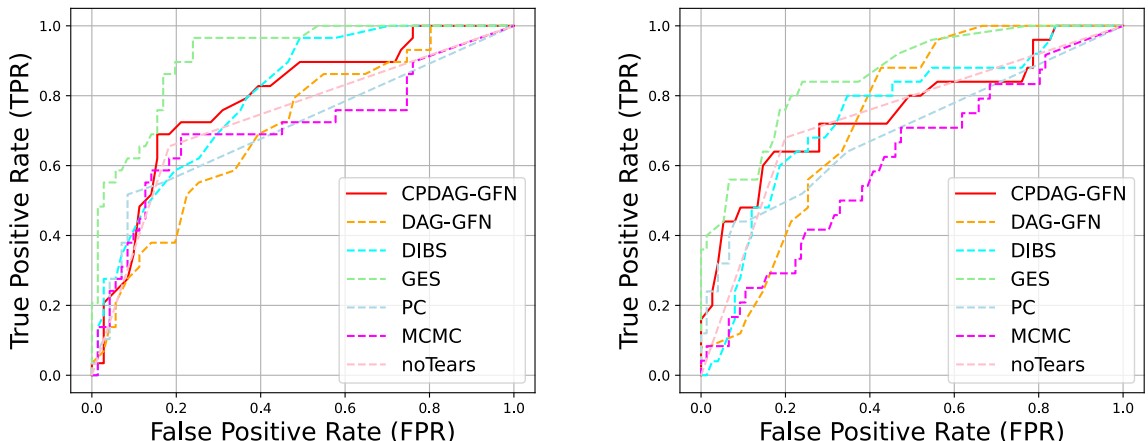

Figure 19: ROC curve comparison for different random seeds under the setting of 100 observational data points, 0.1 noise, and SF-deg3.

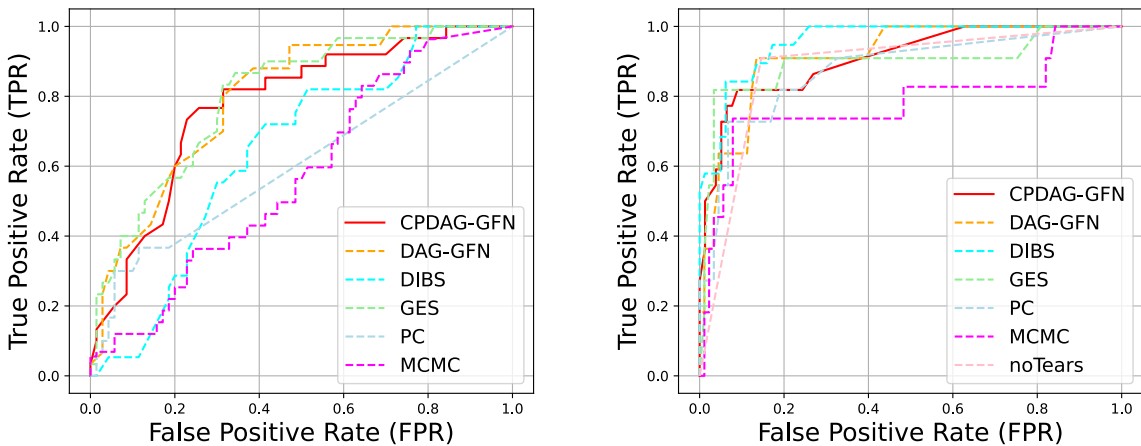

Figure 20: ROC curve comparison for different random seeds under the setting of 100 observational data points, 0.1 noise, and ER-deg2.

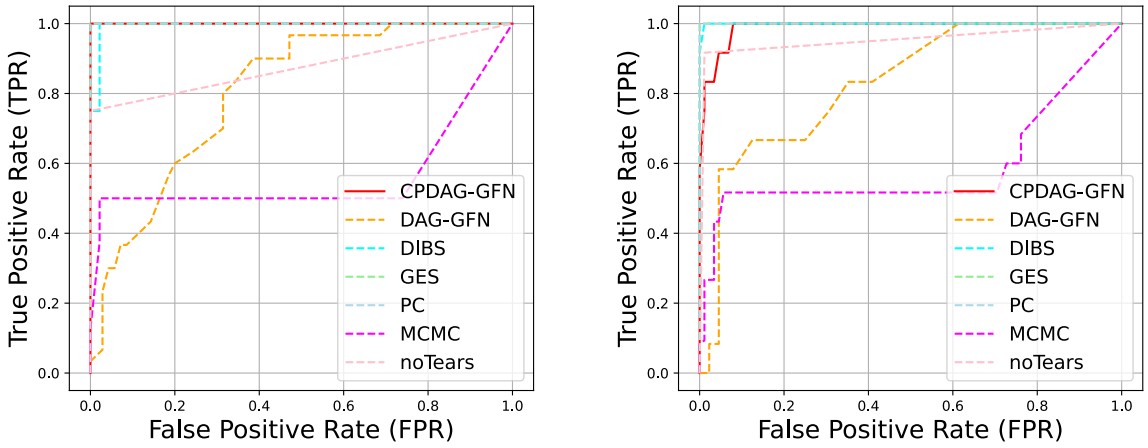

Figure 21: ROC curve comparison for different random seeds under the setting of a Million observational data points, 0.1 noise, and ER-deg1.

## J    Further Analysis

Recall that CPDAG-GFN consists of two steps: (1) using GFlowNet as an amortized sampler to sample top-K high-scoring CPDAGs, and (2) applying a heuristic filter to refine the candidate set by removing the least common edges. Both steps are integral to the method. In this section, we explore what may be causing CPDAG-GFN to perform competitively compared to other baselines. First, we examine the effect of the heuristic filter by comparing the top-K sampled CPDAGs before and after the filter is applied. Our results in Figure 22 show that applying the heuristic filter to the top-K samples from GFlowNet significantly reduces E-SHD, improving the alignment of the sampled graphs with the ground truth. This improvement allows CPDAG-GFN to achieve competitive performance against other baselines.

Next, a natural question arises: Is CPDAG-GFN's competitive performance due to the heuristic filter alone (i.e., would applying the same filter to other baselines yield similar results?), or does the filtering mechanism

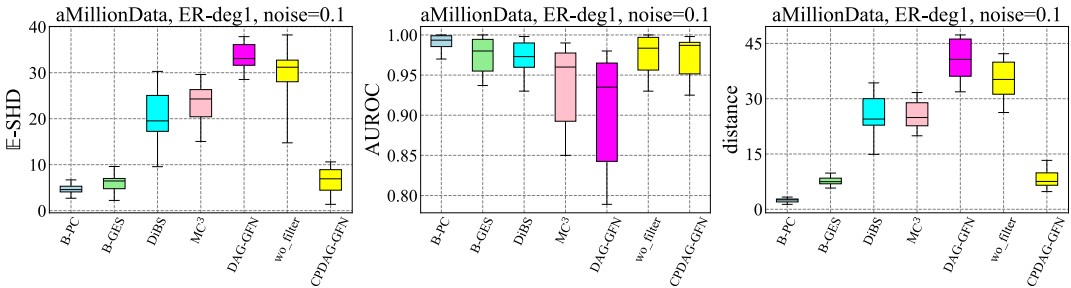

Figure 22: A comparison of baselines where filter is applied and before it is applied. Higher AUROC and lower E-SHD are preferred.

work particularly well when applied to the top-K samples from GFlowNet? To investigate this, we applied the heuristic filter to all baselines.

Our results in Figure 23 show that while the heuristic filter improves performance for the baselines, the improvement is not as substantial as in CPDAG-GFN. Specifically, while the heuristic filter leads to some reduction in E-SHD for baselines like MC3, DAG-GFN, and DiBS, CPDAG-GFN, in particular, becomes more competitive with methods B-PC and B-GES after applying the filter.

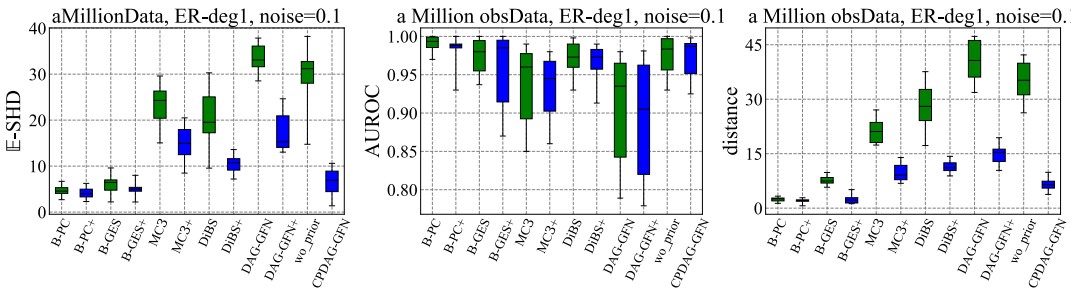

Figure 23: A comparison of baselines where before heuristic filter is applied. Baselines with a plus sign (e.g. blue box) indicate the application of heurisitc filter, and green box is without. Please note that wo-prior means CPDAG-GFN without heuristic filter applied. Higher AUROC and lower E-SHD are preferred.

Possible reasons why the heuristic filter is less effective for other baselines:

**Sampling Strategy:** DAG-GFN, DiBS, and MC3 draw samples from a learned posterior distribution and so are not guaranteed to always sample high-scoring graphs. As a result, their samples may still include a mix of high-, moderate-, and occasionally low-scoring graphs. In contrast, CPDAG-GFN explicitly prioritizes high-scoring graphs by using GFlowNet as an amortized sampler to sample the top K high-scoring CPDAGs. These top-K graphs tend to share common edges, many of which align with the ground truth. This structural consistency allows the heuristic filter to effectively refine the candidate set by removing the least common edges, thereby improving alignment with the true structure. However, when sampling does not explicitly prioritize high-scoring graphs but instead draws from a learned posterior distribution, the heuristic filter appears less effective in Figure 23 in reducing E-SHD compared to CPDAG-GFN.

**Graphs are already similar and close to ground truth:** bootstrapping-PC/GES have low distance as shown in the distance plot. This suggests the sampled graphs are already similar to each other, and these graphs are in good alignment with the ground truth (as indicated by low E-SHD and high AUROC). Since these graphs already share a strong structural resemblance to the ground truth, the heuristic filter has limited room for refinement, resulting in less substantial improvement compared to CPDAG-GFN.

These experimental results demonstrate that the heuristic filter alone is insufficient to achieve strong performance; CPDAG-GFN's competitive performance arises from the synergy between using GFlowNet as an amortized sampler to sample top-K graphs and applying the heuristic filter.

# K  Distinction Between the CPDAG-GFN and DAG-GFN Approaches

In this section, we highlight the key differences between CPDAG-GFN and DAG-GFN. While both methods use the GFlowNet framework and have similar names, this is where the similarities end. Our approach diverges in several key aspects as outline below:

**1. Different Learned Posterior Distribution:**
DAG-GFN uses GFlowNet to learn a posterior distribution over DAGs. In contrast, CPDAG-GFN learns a posterior distribution over equivalence classes of DAGs (i.e., CPDAGs).

**2. Different Use of the GFlowNet Posterior:**
DAG-GFN relies on the learned posterior to produce a set of candidate graphs where the samples are drawn at random from this distribution. As a result, the sampled graphs may span a wide range of scores, including high-, moderate-, and low-scoring candidates. In contrast, CPDAG-GFN leverages the posterior as an amortized sampler, allowing the method to prioritize high-scoring graphs.

**3. Different Search Space and Action Space**

- **DAG-GFN:** Operates in the space of DAGs, where the primary constraint is to avoid generating cycles when adding directed edges (i.e., the action space consists only of adding directed edges).

- **CPDAG-GFN:** The action space consist of adding directed edges, adding undirected edges, and creating v-structures ("makeV"). Each valid action must satisfy all four properties outlined in Theorem 1 of our paper. As a result, CPDAG-GFlownet must account for additional structural rules beyond simple cycle prevention.

**4. Different Neural Network Architecture**

- The graph neural network architecture of DAG-GFN, which handles only directed edges, employs a linear transformer. While efficient, the linear transformer may limit its capacity to capture complex, non-linear dependencies in observational data.

- In contrast, we used Relational Graph Convolutional Network (RGCN) (see section 3.2.3). Our choice of RGCN allows us to effectively model all three edge types (undirected, directed, and v-structures) in CPDAGs, capturing more complex relationships and structural properties in the data. This makes our implementation fundamentally different from that of DAG-GFN.

- Additionally, our decoder differs from that of DAG-GFN. We use SimplE, which we found provides better performance compared to the decoder used in DAG-GFN.

**5. Heuristic Filter** Our method introduces a heuristic filter that improves the quality of the sampled CPDAGs. This component is absent in DAG-GFN, where only a uniform prior is used. We found that relying solely on a uniform prior was not sufficient for producing competitive results in our experimental settings, motivating the development of the filter.

## L    Results for CPDAG-GFN on Larger Networks (20 and 50 Variables)

**Results for d = 20 variables**

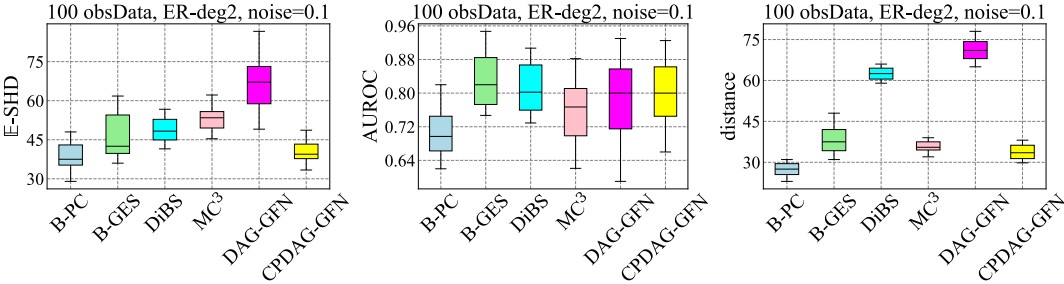

Figure 24: Comparison of E-SHD and AUROC metrics on a dataset with 100 observations, generated using ground truth graphs sampled from an Erdős-Rényi (ER-deg2) model with a noise level of 0.1. Lower E-SHD and higher AUROC are preferred. A higher distance in the third figure indicates greater dissimilarity among the graphs.

**Results for d = 50 variables**

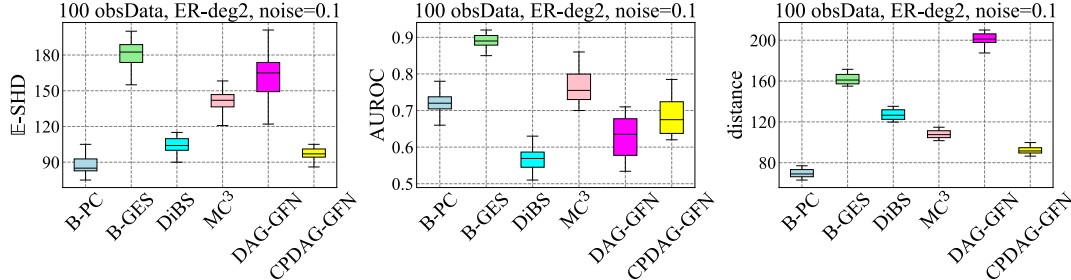

Figure 25: Comparison of E-SHD and AUROC metrics on a dataset with 100 observations, generated using ground truth graphs sampled from an Erdős-Rényi (ER-deg2) model with a noise level of 0.1. Lower E-SHD and higher AUROC are preferred. A higher distance in the third figure indicates greater dissimilarity among the graphs.

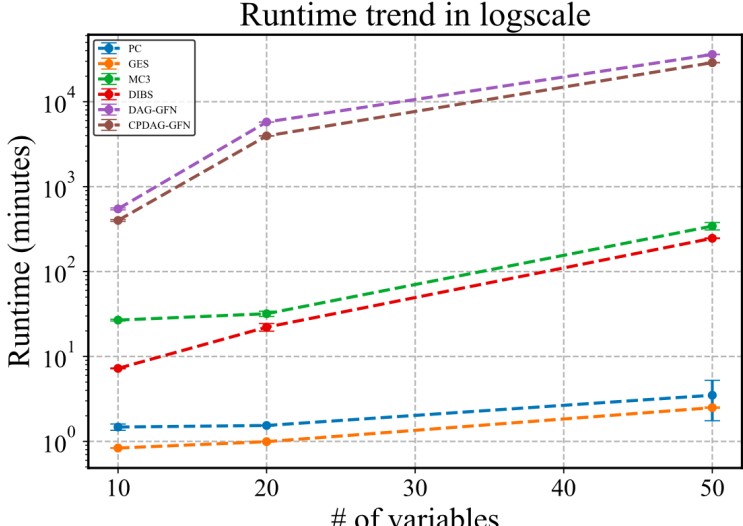

Figure 26: Runtime trends in log-scale with 100 observational samples, noise level 0.1, and 2 EPN.

## M  Runtime trend

We report the runtime trend for each method in Figure 26. All experiments were run on an Apple M4 Pro CPU (12 cores) with 24GB unified memory. As expected, bootstrapped PC and bootstrapped GES run fastest: although bootstrapping adds resampling overhead, the underlying PC and GES procedures are point-estimate methods, which are inherently faster than Bayesian approaches. CPDAG-GFN runs slower than DiBS and MC³, primarily due to the overhead of training a generative flow network. While our method traverses CPDAG space, this factor alone does not explain the slowdown — as shown by the fast runtime of bootstrapped GES, which also operates in the CPDAG space.

A second factor is that our current implementation has not been optimized for runtime efficiency, as this lies outside the scope of the present work; however, this does not affect the methodological contribution of the work. Importantly, the observed trends align with expectations across baselines: bootstrapped point-estimate methods are faster, while Bayesian approaches trade speed for posterior modeling. We also note that GFlowNets represent a different class of learning-based methods whose cost may amortize as problem complexity and scale increase, which may offer more favorable scalability in larger domains.

## N  Qualitative analysis of BayesDAG and BCDNets

In this section, we provide a qualitative comparison between two recent methods in Bayesian causal discovery: BCDNets and BayesDAG.

In 2021, Cundy et al. (2021) proposed Bayesian Causal Discovery Nets (BCDNets), a variational inference framework for estimating a distribution over DAGs that characterize a linear-Gaussian structural equation model (SEM). A key strength of BCDNets lies in its use of an expressive variational family of factorized posterior distributions over the SEM parameters and continuous relaxations for optimization, enabling scalability to high-dimensional settings Cundy et al. (2021). One of the drawbacks with their method is their strong assumption that the true data-generating process follows a linear-Gaussian SEM. While this assumption is common in causal discovery, the true data-generating process may not always follow a linear-Gaussian SEM in practice, constituting a strong modeling assumption that may limit the applicability of BCDNets in settings where the underlying process deviates from a linear-Gaussian SEM. In contrast, CPDAG-GFN avoids assuming a specific functional form for the data-generating process and learns a distribution over equivalence classes of graphs (CPDAGs), which enables direct sampling of CPDAGs.

The second drawback of BCDNets is that it relies solely on variational inference, which may lead to compromised inference accuracy (Annadani et al., 2023). In response to this limitation and the one mentioned above, Annadani et al. (2023) introduced BayesDAG, a scalable Bayesian causal discovery framework that combines stochastic gradient Markov Chain Monte Carlo (SG-MCMC) and variational inference (VI) to enable sampling directly from the posterior distribution over DAGs. One key strength of BayesDAG is its applicability not only to linear causal relationships but also to nonlinear ones. This gives BayesDAG greater flexibility than BCDNets when modeling datasets where linearity assumptions may not hold.

Experimental results from Experimental results from Annadani et al. (2023) show that BCDNets performs competitively with BayesDAG on Erdős–Rényi graphs, and outperforms BayesDAG on scale-free graphs under linear-Gaussian data settings for a small variable space of d=5. This raises the question of how the performances of these two methods compare when scaling to larger variable spaces.

Although BayesDAG aims to improve inference quality by combining SG-MCMC and VI, this design choice raises questions about the computational complexity and potential trade-offs compared to purely variational methods like BCDNets in linear data settings. Specifically, while BCDNets may introduce inference errors due to its reliance on variational inference alone, BayesDAG's use of SG-MCMC may present challenges commonly associated with MCMC methods, such as slow mixing and uncertainty in determining convergence. In practice, assessing convergence can be difficult, and early samples may bias the inferred causal structures. Nonetheless, BayesDAG demonstrates a clear advantage over BCDNets on nonlinear synthetic datasets, highlighting its greater flexibility in modeling more complex causal relationships.

