# OpenReview forum: "Learning Equivalence Classes of Bayesian Network Structures with GFlowNet"
_TMLR — Accepted by TMLR_

### Review · Reviewer_tz9G · 2025-04-11

**Summary Of Contributions:**

The authors address the problem of learning equivalence classes of Bayesian network structures from observational data using Generative Flow Networks (GFlowNets). Their approach involves learning a posterior distribution over Completed Partially Directed Acyclic Graphs (CPDAGs) and identifying an optimal set of $K$ candidates that achieve the highest scores from the amortized sampling. To refine the results, the authors perform postprocessing by removing the least common edges among the sampled graphs. The proposed GFlowNet framework includes a state space represented by CPDAGs, a reward function, and a parameterization with graph neural networks. The authors validate their method through numerical experiments conducted on both real and synthetic datasets, demonstrating its applicability and effectiveness.

**Audience:**

Yes

**Broader Impact Concerns:**

N/A.

**Claims And Evidence:**

Yes

**Requested Changes:**

Please see the previous section. Additionally, I also sighted a few minor writing issues in the paper:
1. I could not find the definition of the abbreviation MLP used on page 6.
2. I suspect that DAG-GFlowNet and DAG-GFN are used interchangeably on page 2.
3. In Figure 8, DAG+ is perhaps used to denote the use of the filter. However, this is not made clear in the text.
4. Appendix E has a missing Figure reference in line 3.
5. Figure 22 has repeated labels (y-axis) for two of the figures.

**Strengths And Weaknesses:**

**Strength:**

The structure learning of Bayesian networks (up to their equivalence class) is an important problem in machine learning. Rather than learning a posterior over DAGs directly, the authors propose a method that uses GFlowNet to learn a posterior over CPDAGs. The two important contributions come in terms of a) yielding Top-$K$ scoring CPDAGs and b) proposing a new filtering heuristic. The paper is generally well-written and conveys its ideas clearly.

**Weaknesses:**

1. The paper borrows heavily from the work by Deleu et al. (2022), adopting the key idea of employing GFlowNet while introducing modifications such as utilizing a state space of CPDAGs instead of DAGs and incorporating a Top-K approach with filtering. However, the authors could strengthen their contribution by clearly delineating other technical distinctions from Deleu et al. (2022) and elaborating on the specific technical challenges encountered in adapting and extending these methods to their proposed setup.
2. Given the limited technical novelty, the primary value of the paper could have been demonstrated through superior performance against existing baselines across a diverse range of examples or performance metrics. Unfortunately, the numerical results presented are only competitive, with performance varying across different metrics. Notably, some of the most intriguing aspects of the paper are relegated to the Appendix (specifically sections D, E, F, H, I, and J). Upon reviewing the provided figures, it appears that DAG-GFN from Deleu et al. (2022), when equipped with Top-K sampling and filtering heuristics, performs comparably to the proposed method.
3. I am unclear about the process for selecting hyperparameters such as K and L. While Appendix E outlines an approach for choosing L, it is not evident whether the suggested range of 30–40 for L would remain valid across Bayesian networks with differing numbers of nodes or edges. Authors should also specify these numbers in the experiments they reported.
4. It would be a good addition to have experiments showing whether the proposed method scales with the number of nodes/edges in the Bayesian networks. Authors should clearly specify these numbers in the experiments they reported.

---

> ### Author Response · Authors · 2025-05-02
> **response to Reviewer tz9G**
>
> >1. The paper borrows heavily from the work by Deleu et al. (2022), adopting the key idea of employing GFlowNet while introducing modifications such as utilizing a state space of CPDAGs instead of DAGs and incorporating a Top-K approach with filtering. However, the authors could strengthen their contribution by clearly delineating other technical distinctions from Deleu et al. (2022) and elaborating on the specific technical challenges encountered in adapting and extending these methods to their proposed setup.
>
> Thank you for your feedback. It appears we did not make the distinction between DAG-GFN and CPDAG-GFN sufficiently clear in the paper, which may have led to this impression. To address this, we have added Appendix K to the manuscript. Please refer to the updated manuscript for details.
>
> >2. the numerical results presented are only competitive, with performance varying across different metrics.
>
> Thank you for your feedback. AUROC and E-SHD capture different aspects of performance and are best interpreted as complementary rather than in isolation. A method that performs well on both metrics (e.g., achieving lower E-SHD and higher AUROC) generally demonstrates stronger overall performance than one that excels at only one. CPDAG-GFN performs competitively, often matching or exceeding established baselines (particularly DAG-GFN).
>
> >2. Upon reviewing the provided figures, it appears that DAG-GFN from Deleu et al. (2022), when equipped with Top-K sampling and filtering heuristics, performs comparably to the proposed method.
>
> We appreciate the reviewer’s observation and would like to clarify that the Top-K sampling and filtering heuristics are a variant of our method adapted to the DAG space.  TopK-DAG is a baseline we introduced in the appendix to analyze how key components of CPDAG-GFN—specifically amortized sampling and the heuristic filter—perform when applied in the DAG space. As shown in Appendix D, TopK-DAG outperforms DAG-GFN. We have revised Appendices D and F in the updated manuscript to clarify topK-DAG.
>
> >3. I am unclear about the process for selecting hyperparameters such as K and L. While Appendix E outlines an approach for choosing L, it is not evident whether the suggested range of 30–40 for L would remain valid across Bayesian networks with differing numbers of nodes or edges. Authors should also specify these numbers in the experiments they reported.
>
> Thank you for your comments. The parameter K is determined by the user and reflects the number of CPDAG candidates the model is requested to output. The hyperparameter L can be tuned using standard hyperparameter tuning methods such as grid search.
>
> In general, the appropriate value for L will vary depending on the Bayesian network’s number of nodes and average degree. For example,  a large network with 50 nodes with an average degree of 30 will likely require a different value of L compared to a small sparse network with 3 nodes and an average degree of 1.  We will specify the L values in the revised manuscript.
> For 10-node networks in our main experiments, we used L = 35 for Erdős–Rényi graphs with an expected degrees of 1 and 2, and L = 40 for scale-free graph with an average expected degree of 3.
>
> >4. It would be a good addition to have experiments showing whether the proposed method scales with the number of nodes/edges in the Bayesian networks. Authors should clearly specify these numbers in the experiments they reported.
>
> Thank you for your comment. Conducting scalability experiments is computationally expensive, particularly in terms of memory and GPU resources. We also face constraints in computational resources, with limited memory and restricted GPU access. For graphs larger than 50 variables, we encountered memory bottlenecks, and one of the baselines ran out of memory when processing graphs with 60 variables. We were only able to conduct experiments with up to 50 variables. To provide some preliminary insights into how our method scales with the number of nodes, we have included results for networks with 20 and 50 variables in Appendix L. We acknowledge the importance of scalability analysis and consider it a valuable direction for future research.

---

> ### Author Response · Authors · 2025-05-02
> **further response to requested changes**
>
> >5.1. I could not find the definition of the abbreviation MLP used on page 6.
>
> We thank the reviewer for pointing this out. We have now defined the abbreviation MLP (multilayer perceptron) in the updated manuscript.
>
> >5.2. I suspect that DAG-GFlowNet and DAG-GFN are used interchangeably on page 2.
>
> Thank you for pointing this out. We have updated the manuscript to consistently use "DAG-GFN" instead of "DAG-GFlowNet".
>
> >5.3. In Figure 8, DAG+ is perhaps used to denote the use of the filter. However, this is not made clear in the text.
>
> Thank you for pointing this out. We have edited the manuscript to clarify the meaning of DAG+ in Figure 8 by adding the following sentence: "Baselines with a plus sign indicate that the heuristic filter has been applied."
>
> >5.4. Appendix E has a missing Figure reference in line 3.
>
> Thank you for pointing this out. We fixed it and have updated the manuscript.
>
> >5.5 Figure 22 has repeated labels (y-axis) for two of the figures.
>
> Thank you for the feedback, we fixed the issue and updated the manuscript accordingly.

---

### Review · Reviewer_MSdv · 2025-04-21

**Summary Of Contributions:**

This work introduces a model for modeling a posterior distribution over the completed partially directed acyclic graph (CPDAGs) given observational data. To achieve this, the paper builds on GflowNets to learn a model that uses graph actions to sample the CPDAG sequentially. The authors introduce a sample based edge filter that removes the least common edges, which improves performance of the method by removing spurious edges. The method is evaluated on both synthetic datasets (across varying sample sizes, noise levels, and graph topologies) and a real-world protein signaling dataset (Sachs et al., 2005). Results are reported using the Expected Structural Hamming Distance (E-SHD), AUROC, and edge count. CPDAG-GFN performs competitively with, and in many cases outperforms, state-of-the-art baselines including DAG-GFN, DiBS, and MCMC-based methods.

**Audience:**

Yes

**Claims And Evidence:**

Yes

**Requested Changes:**

- All experiments are performed on relatively small graphs with 10 variables. It is unclear how CPDAG-GFN scales to more realistic settings. The authors should provide either a larger study (50-100) or a justification of what are the scalability issues (lack of convergence or runtime / memory issues?).
- I would like the authors to assess the E-CPDAG SHD as it’s more suitable for the task.
- While the method is compared to strong baselines such as DAG-GFN, DiBS, and MC3, it omits evaluation against BayesDAG and BCDNets, which are recent and competitive Bayesian approaches to structure learning. These methods have shown strong empirical performance in both low- and high-dimensional settings. A comparison—either empirical or in-depth qualitative—would help better position CPDAG-GFN in the current landscape of causal discovery algorithms.
- Figure 4 titles have some issues (e.g. aMillionData vs a Million obs Data - please unify)

**Strengths And Weaknesses:**

# Strengths:

- The authors propose a principled and well-motivated approach that directly samples CPDAGs rather than DAGs, addressing the core challenge that causal discovery from observational data is only identifiable up to a Markov Equivalence Class (MEC). This design decision avoids unnecessary conversions and leads to a more faithful exploration of the relevant search space.
- The paper proves that the selected set of actions (add directed/undirected edge, make-v) is sufficient to construct any valid CPDAG. This theoretical grounding strengthens the design of the search space and provides clarity on completeness.
- The proposed filtering method for removing least-common edges among sampled CPDAGs is simple and effective. Empirical evidence shows that it leads to lower E-SHD and better alignment with ground truth, particularly when combined with GFlowNet’s top-K sampling.
- Quantitatively, CPDAG-GFN outperforms or matches the best-performing baselines (e.g., DiBS, DAG-GFN, PC, GES) across most settings in terms of E-SHD, AUROC, and closeness to the correct number of edges.

# Weaknesses

- While the edge-sparsity heuristic works well empirically, it is not grounded in a formal Bayesian or statistical framework. The lack of theoretical understanding of when and why it works may limit its generalization to other domains or variants of the problem.
- Although experiments are thorough, most settings are limited to 10-variable graphs. It would be helpful to see performance and runtime trends for higher-dimensional settings (e.g., 20–50 variables), which are more common in practical applications.
- The model assumes a uniform prior over CPDAGs, which simplifies training but ignores the possibility of incorporating domain-specific structure priors. This could be particularly limiting in applications (e.g., genomics, medicine) where partial prior knowledge is available.
- The evaluation focuses on graph recovery metrics (E-SHD, AUROC), but does not assess downstream causal tasks such as interventional prediction or counterfactual reasoning. It would strengthen the paper to include at least one such experiment.
- The authors missed to compare with baselines such as or BCDNets [2021] or BayesDAG [ 2023]
- This work uses E-SHD although a more suitable metric exist that evaluates CPDAGs (E-CPDAG SHD) (see "Challenges and considerations in the evaluation of Bayesian causal discovery” and “BayesDAG: Gradient-Based Posterior Inference for Causal Discovery”)

---

> ### Author Response · Authors · 2025-05-02
> **response to Weaknesses**
>
> >While the edge-sparsity heuristic works well empirically, it is not grounded in a formal Bayesian or statistical framework. The lack of theoretical understanding of when and why it works may limit its generalization to other domains or variants of the problem.
>
> Thank you for your comment. While our heuristic is not grounded in a formal Bayesian framework, we acknowledge this limitation and agree that a theoretical analysis would further strengthen our understanding. As a first step toward assessing generalizability, we include empirical evidence in Appendix F and further analysis in Appendix J, where we apply the same heuristic filtering to other baselines. The results show that the heuristic improves their performance as well—though the gains are most pronounced when applied to top-K samples from the GFlowNet. Nevertheless, this suggests the approach may extend to related methods, particularly those that generate diverse high-reward samples. We view this as an initial indication of broader applicability, while recognizing that a formal theoretical grounding remains an important direction for future work.
>
> >Although experiments are thorough, most settings are limited to 10-variable graphs. It would be helpful to see performance and runtime trends for higher-dimensional settings (e.g., 20–50 variables), which are more common in practical applications.
>
> Thank you for the feedback. We have included additional experiments on higher-dimensional settings (20 and 50 variables) in Appendix L of the updated manuscript.
>
> >The model assumes a uniform prior over CPDAGs, which simplifies training but ignores the possibility of incorporating domain-specific structure priors. This could be particularly limiting in applications (e.g., genomics, medicine) where partial prior knowledge is available
>
> Thank you for the feedback. We agree that incorporating domain-specific structure priors could be valuable in settings such as genomics or medicine. However, our current work focuses on the uniform prior setting, where we assume no prior knowledge is available. As noted in the conclusion, extending CPDAG-GFN to incorporate domain-specific priors is part of our future work.
>
> >The evaluation focuses on graph recovery metrics (E-SHD, AUROC), but does not assess downstream causal tasks such as interventional prediction or counterfactual reasoning. It would strengthen the paper to include at least one such experiment.
>
> Thank you for your feedback. We agree that evaluating downstream causal tasks such as interventional prediction and counterfactual reasoning is valuable, particularly in broader causal inference pipelines. However, in this work, our primary goal is to improve causal discovery by generating diverse CPDAG candidates that capture structural uncertainty from observational data, and we chose to keep the scope focused on this core contribution rather than extending into downstream tasks, which we view as complementary to the current work.

---

> ### Author Response · Authors · 2025-05-02
> **response to requested changes**
>
> >All experiments are performed on relatively small graphs with 10 variables. It is unclear how CPDAG-GFN scales to more realistic settings. The authors should provide either a larger study (50-100) or a justification of what are the scalability issues (lack of convergence or runtime / memory issues?).
>
> Thank you for your feedback. We have added additional experiments for 50 variables in Appendix L (please see the updated manuscript). Due to resource constraints — particularly limited memory availability and restricted GPU access, we were only able to conduct experiments up to 50 variables. Memory bottlenecks arise when running both our method and the baselines on graphs larger than 50 variables. Specifically, we observed that one of the baselines ran out of memory for graphs with 60 variables. Additionally, training times increase as graph size grows for both the baselines and our method. While GPU resources are available in our environment, their usage is restricted, and long-running jobs are subject to preemption, which makes large-scale experiments challenging.
>
>
> > I would like the authors to assess the E-CPDAG SHD as it’s more suitable for the task.
>
> We thank the reviewer for the helpful suggestion regarding the evaluation metric. Upon reviewing the definitions of E-SHD and E-CPDAG SHD in the references provided, particularly ref [1], we find that our metric E-SHD as defined in Appendix B, corresponds to E-CPDAG SHD. We summarize the key difference below.
>
> In ref [1], E-SHD is defined differently, where SHD measures the number of edges to be added, removed, or reversed to match the ground truth graph. In contrast, our SHD definition in Appendix B is stated as:  ``The SHD($G_k, G^*$) counts the number of edge changes that separate the learned CPDAGs $G_k$ from the ground truth $G^*$.''  Thus, our evaluation computes the expected SHD between CPDAGs, consistent with the definition of E-CPDAG SHD in ref [1].
>
>
> References:
>
> [1] Mamaghan, A. M. K., Tigas, P., Johansson, K. H., Gal, Y., Annadani, Y., & Bauer, S. (2024). Challenges and considerations in the evaluation of Bayesian causal discovery. arXiv preprint arXiv:2406.03209.
>
> > While the method is compared to strong baselines such as DAG-GFN, DiBS, and MC3, it omits evaluation against BayesDAG and BCDNets, which are recent and competitive Bayesian approaches to structure learning. These methods have shown strong empirical performance in both low- and high-dimensional settings. A comparison—either empirical or in-depth qualitative—would help better position CPDAG-GFN in the current landscape of causal discovery algorithms.
>
> Thank you for the suggestion. In response, we have added an in-depth qualitative comparison in Appendix M. Please refer to the updated manuscript for details.
>
> > Figure 4 titles have some issues (e.g. aMillionData vs a Million obs Data - please unify)
> Done! thank you for pointing that out.

---

### Review · Reviewer_naA1 · 2025-04-23

**Summary Of Contributions:**

This paper introduces CPDAG-GFN, a novel algorithm that directly samples Completed Partially Directed Acyclic Graphs (CPDAGs) using Generative Flow Networks. By operating in CPDAG space rather than DAG space, the method efficiently generates multiple high-scoring equivalence-class candidates without the extra conversion step.

**Audience:**

Yes

**Claims And Evidence:**

Yes

**Requested Changes:**

- In Section 3.2.1, “… ensuring that the graph resulting from any permitted action will also be a CDPAG.” Should read “CPDAG.”
- As suggested in previous reviews, please add a brief paragraph on differentiable DAG learning methods (e.g., NOTEARS) and move the corresponding experimental comparisons into the main text.
- Figure 3 nicely illustrates the CPDAG construction process, but it remains unclear how transitions between states are chosen. Are actions sampled via the learned forward policy, weighted by reward, or exhaustively enumerated? Please elaborate on the mechanism and whether all feasible actions are considered at each step.
- The paper argues that “relying on the score to prioritize graphs may lead to discrepancies with the ground truth.” However, score-based DAG learning typically assumes that the true DAG obtains the highest score under a correct scoring function. Could the authors clarify why, in this setting, a high score might diverge from the ground truth?

**Strengths And Weaknesses:**

### **Strengths**

- The paper presents a fresh integration of GFlowNets into causal discovery by learning CPDAGs directly, which is timely given the rising interest in GFlowNet methods.
- Working in CPDAG space eliminates the need to convert sampled DAGs into CPDAGs, focusing the sampler on equivalence classes and improving efficiency.

### **Weaknesses**

- As GFlowNets may be unfamiliar to some readers, it would help to include a boxed “Algorithm” in the main text that lays out the full CPDAG-GFN procedure in a single, coherent framework.

---

> ### Author Response · Authors · 2025-05-02
> **response to weaknesses and requested Change**
>
> >As GFlowNets may be unfamiliar to some readers, it would help to include a boxed “Algorithm” in the main text that lays out the full CPDAG-GFN procedure in a single, coherent framework.
>
> Thank you for your thoughtful suggestion to include a boxed “Algorithm” in the main text to present the full CPDAG-GFN procedure. We agree that such summaries can often improve clarity. Since our method closely follows the standard GFlowNet framework—which is already well-documented in the literature and tutorials—we opted to describe only the CPDAG-specific components in detail in the manuscript. These include but not limited to  reward and state space which are the core elements required to train GFlowNet. That said, we are open to including a boxed algorithm in the final manuscript if the reviewer still prefers it.
>
> Requested Change
>
> > In Section 3.2.1, “… ensuring that the graph resulting from any permitted action will also be a CDPAG.” Should read “CPDAG.”
>
> Thank you for pointing that out. We have corrected the typo in Section 3.2.1 and updated the manuscript accordingly.
>
> >"As suggested in previous reviews, please add a brief paragraph on differentiable DAG learning methods (e.g., NOTEARS) and move the corresponding experimental comparisons into the main text."
>
> Thank you for the suggestion. We have added a brief paragraph on NOTEARS in Appendix X of the updated manuscript. We would respectfully ask whether you would reconsider the request to move NOTEARS into the main text. Our reasoning is as follows: we included NOTEARS in the appendix rather than in the main experimental section because its modeling assumptions and evaluation protocol differ substantially from ours and from those of the other baselines. Specifically, NOTEARS is a point-estimate method that returns a single DAG, and its output is sensitive to the choice of a threshold parameter. As a result, its evaluation is inherently threshold-dependent—unlike AUROC, which reflects ranking performance across all thresholds. Thus, AUROC is particularly well suited for posterior-sampling methods, as adopted in our work and in prior literature, but is less appropriate for evaluating methods like NOTEARS. Moreover, our method and all main baselines sample from a posterior distribution, whereas NOTEARS does not. Consequently, the distance metric reported in the main text is not applicable to NOTEARS.  Given these differences in modeling assumptions and evaluation objectives, we believe it is more appropriate to include NOTEARS in the appendix rather than as a core baseline in the main text.   That said, we are open to discussing this further if the reviewer feel strongly about including NOTEARS in the main text.
>
>
> > Figure 3 nicely illustrates the CPDAG construction process, but it remains unclear how transitions between states are chosen. Are actions sampled via the learned forward policy, weighted by reward, or exhaustively enumerated? Please elaborate on the mechanism and whether all feasible actions are considered at each step.
>
> Thank you for your feedback and question. At each state, an action is sampled from the learned forward policy $P(G_{i+1} \mid G_i, a_i)$, where \$G_{i+1} $ represents the new state, $G_i$ represents the current state, and $ a_i $ is the action that takes you from the current state $ G_i $ to the next state. Actions are neither weighted by reward nor exhaustively enumerated. Instead, the model stochastically samples one action from the learned forward policy to transition to the next state (CPDAG). Moreover, to ensure that only valid actions are considered, a mask is used to filter out all invalid actions. As a result, the forward policy assigns essentially zero probability to sampling invalid actions.
>
> In the diagram you referenced, the initial state is a disconnected graph. To transition to a new state, an action is sampled from the forward policy $ P(G_{i+1} \mid G_i, a_i) $, which is derived from neural network. If a stop action is sampled from the policy, the trajectory terminates, and the reward is computed. Otherwise, the sampled action transitions from the current state (e.g., a CPDAG) to a new state (another CPDAG), and this process continues until a stop action is sampled.

---

> ### Author Response · Authors · 2025-05-02
> **further response to requested changes**
>
> > The paper argues that “relying on the score to prioritize graphs may lead to discrepancies with the ground truth.” However, score-based DAG learning typically assumes that the true DAG obtains the highest score under a correct scoring function. Could the authors clarify why, in this setting, a high score might diverge from the ground truth?
>
> Thank you for your comment. We agree that score-based DAG learning generally assumes that the true DAG should obtain the highest score under a correct scoring function. However, there are situations in which the highest-scoring model may not correspond to the true underlying structure. Below are some reasons why a high score might diverge from the ground truth in such settings:
>
> **Noise in the data**: When a model (in this case, a learned graph) is trained on noisy data, it might fit not only to the true underlying relationships but also to the noise itself. This results in the model capturing spurious relationships or patterns that do not exist in the true graph. Since the scoring function used to evaluate the learned graph  reflects how well the graph fits the observed data, a graph that  overfits the noise  can end up achieving a higher score than the ground truth.
>
> **Insufficient Data:** As noted in [1], when there is insufficient data to adequately constrain the model, multiple models may fit the data well. Due to the scarcity of data relative to the number of variables, several models may explain the data reasonably well. In such cases, the highest-scoring model may not necessarily represent the true underlying process (i.e., the true model). The lack of sufficient data limits the model's ability to uniquely identify the true graph, meaning that the highest-scoring graph may not correspond to the true graph.
>
> **No perfect score function:**  Additionally, to the best of our knowledge, there is no perfect score function. By "perfect," we mean a score function that always assigns the highest score to the true graph (out of all possible graphs) across all possible settings.
>
> References:
> [1] Friedman, N., & Koller, D. (2013). Being Bayesian about network structure. arXiv preprint arXiv:1301.3856

---

### Decision · Action_Editor_W4zP · 2025-06-12

**Recommendation:** Accept with minor revision

**Additional Comments:**

Please remove the word "state-of-the-art" from the paper, as the comparison methods (e.g., PC and GES) are hardly considered in such category.

In addition, please include the technique of (Tsamardinos et al 2006, "The max-min hill-climbing Bayesian network structure learning algorithm") in the empirical evaluation. This is a classical well-known method with good performance.

If I understand correctly, (Sachs et al 2025, "Causal Protein-Signaling Networks Derived from Multiparameter Single-Cell Data") did not provide a ground truth, but instead the paper learned a network from data. Thus, assuming the (Sachs et al 2025) network as being a ground truth seems to be an incorrect experimental setup. I suggest using independent objective measures such as test log-likelihood to verify the quality of the results.

As suggested by reviewer MSdv, please include some runtime trends (for the proposed and comparison methods) for different number of nodes (e.g., 10, 20, 50 as in Appendix L) so that one could experimentally see how the computational complexity scales with number of nodes.

Finally, I found a small typo on Page 7: "see appendix ?? for further detail"

**Audience:**

Yes

**Audience Explanation:**

All reviewers agree that the manuscript has an audience. Learning Bayesian network structures is a classical machine learning problem studied since the 1990s. The audience might be relatively smaller when compared to main-trend machine learning problems, but the paper is definitely relevant to at least some individuals in the machine learning community and the TMLR readership.

**Claims And Evidence:**

Yes

**Claims Explanation:**

All reviewers agree that the manuscript presents appropriate evidence for its claims. The paper proposes a generative flow network to learn a posterior distribution over completed partially directed acyclic graphs. Experimental evidence is also provided on synthetic and real world data.